# Implicit Regularization of Accelerated Methods in Hilbert Spaces

**Nicolò Pagliana**
University of Genoa
DIMA & MaLGa
pagliana@dima.unige.it

**Lorenzo Rosasco**
University of Genoa
DIBRIS, MaLGa, IIT & MIT
lrosasco@mit.edu

## Abstract

We study learning properties of accelerated gradient descent methods for linear least-squares in Hilbert spaces. We analyze the implicit regularization properties of Nesterov acceleration and a variant of heavy-ball in terms of corresponding learning error bounds. Our results show that acceleration can provides faster bias decay than gradient descent, but also suffers of a more unstable behavior. As a result acceleration cannot be in general expected to improve learning accuracy with respect to gradient descent, but rather to achieve the same accuracy with reduced computations. Our theoretical results are validated by numerical simulations. Our analysis is based on studying suitable polynomials induced by the accelerated dynamics and combining spectral techniques with concentration inequalities.

## 1 Introduction

The focus on optimization is a major trend in modern machine learning, where efficiency is mandatory in large scale problems [4]. Among other solutions, first order methods have emerged as methods of choice. While these techniques are known to have potentially slow convergence guarantees, they also have low iteration costs, ideal in large scale problems. Consequently the question of accelerating first order methods while keeping their small iteration costs have received much attention, see e.g. [33]. Since machine learning solutions are typically derived minimizing an empirical objective (the training error), most theoretical studies have focused on the error estimated for this latter quantity. However, it has recently become clear that optimization can play a key role from a statistical point of view when the goal is to minimize the expected (test) error. On the one hand, iterative optimization implicitly bias the search for a solution, e.g. converging to suitable minimal norm solutions [27]. On the other hand, the number of iterations parameterize paths of solutions of different complexity [31].

The idea that optimization can implicitly perform regularization has a long history. In the context of linear inverse problems, it is known as *iterative regularization* [11]. It is also an old trick for training neural networks where it is called early stopping [15]. The question of understanding the generalization properties of deep learning applications has recently sparked a lot of attention on this approach, which has be referred to as implicit regularization, see e.g. [13]. Establishing the regularization properties of iterative optimization requires the study of the corresponding expected error by combining optimization and statistical tools. First results in this sense focused on linear least squares with gradient descent and go back to [6, 31], see also [25] and references there in for improvements. Subsequent works have started considering other loss functions [16], multi-linear models [13] and other optimization methods, e.g. stochastic approaches [26, 18, 14].

In this paper, we consider the implicit regularization properties of acceleration. We focus on linear least squares in Hilbert space, because this setting allows to derive sharp results and working in infinite dimension magnify the role of regularization. Unlike in finite dimension learning bounds are

possible only if some form of regularization is considered. In particular, we consider two of the most popular accelerated gradient approaches, based on Nesterov acceleration [22] and (a variant of) the heavy-ball method [24]. Both methods achieve acceleration by exploiting a so called momentum term, which uses not only the previous, but the previous two iterations at each step. Considering a suitable bias-variance decomposition, our results show that accelerated methods have a behavior qualitatively different from basic gradient descent. While the bias decays faster with the number of iterations, the variance increases faster too. The two effect balance out, showing that accelerated methods achieve the same optimal statistical accuracy of gradient descent but they can indeed do this with less computations. Our analysis takes advantage of the linear structures induced by least squares to exploit tools from spectral theory. Indeed, the characterization of both convergence and stability rely on the study of suitable spectral polynomials defined by the iterates. While the idea that accelerated methods can be more unstable, this has been pointed out in [10] in a pure optimization context. Our results quantify this effect from a statistical point of view. Close to our results is the study in [9], where a stability approach is considered to analyze gradient methods for different loss functions [5].

## 2  Learning with (accelerated) gradient methods

Let the input space $\mathcal{X}$ be a separable Hilbert space (with scalar product $\langle \cdot, \cdot \rangle$ and induced norm $\|\cdot\|$) and the output space be $\mathbb{R}$ [1]. Let $\rho$ be a unknown probability measure on the input-output space $\mathcal{X} \times \mathbb{R}$, $\rho_{\mathcal{X}}$ the induced marginal probability on $\mathcal{X}$, and $\rho(\cdot|x)$ the conditional probability measure on $\mathbb{R}$ given $x \in \mathcal{X}$. We make the following standard assumption: there exist $\kappa > 0$ such that

$$\langle x, x' \rangle \leq \kappa^2 \qquad \forall x, x' \in \mathcal{X}, \ \rho_{\mathcal{X}}\text{-almost surely.} \tag{1}$$

The goal of least-squares linear regression is to solve the *expected risk minimization* problem

$$\inf_{w \in \mathcal{X}} \mathcal{E}(w), \quad \mathcal{E}(w) = \int_{\mathcal{X} \times \mathbb{R}} (\langle w, x \rangle - y)^2 \, d\rho(x, y), \tag{2}$$

where $\rho$ is known only through the $n$ i.i.d. samples $(x_1, y_1), \ldots, (x_n, y_n)$. In the following, we measure the quality of an approximate solution $\hat{w}$ with the excess risk

$$\mathcal{E}(\hat{w}) - \inf_{\mathcal{X}} \mathcal{E} \ .$$

The search of a solution is often based on replacing (2) with the empirical risk minimization (ERM)

$$\min_{w \in \mathcal{X}} \hat{\mathcal{E}}(w), \qquad \hat{\mathcal{E}}(w) = \frac{1}{n} \sum_{i=1}^{n} (\langle w, x_i \rangle - y_i)^2 \ . \tag{3}$$

For least squares an ERM solution can be computed in closed form using a direct solver. However, for large problems, iterative solvers are preferable and we next describe the approaches we consider.

First, it is useful to rewrite the ERM with vectors notation. Let $\mathbf{y} \in \mathbb{R}^n$ with $(\mathbf{y})_i = y_i$ and $\mathrm{X} : \mathcal{X} \to \mathbb{R}^n$ s.t. $(\mathrm{X}\,w)_i = \langle w, x_i \rangle$ for $i = 1 \ldots, n$. Here the norm $\|\cdot\|_n$ is norm in $\mathbb{R}^n$ multiplied by $1/\sqrt{n}$. Let $\mathrm{X}^* : \mathbb{R}^n \to \mathcal{X}$ be the adjoint of X defined by $\mathrm{X}^* \, \mathbf{y} = \frac{1}{n} \sum_{i=1}^{n} x_i y_i$. Then, ERM becomes

$$\min_{w \in \mathcal{X}} \hat{\mathcal{E}}(w) = \|\mathrm{X}\,w - \mathbf{y}\|_n^2 \ . \tag{4}$$

### 2.1  Gradient descent and accelerated methods

Gradient descent serves as a reference approach throughout the paper. For problem (4) it becomes

$$\hat{w}_{t+1} = \ \hat{w}_t - \alpha\,\mathrm{X}^*\,(\mathrm{X}\,\hat{w}_t - \mathbf{y}) \tag{5}$$

with initial point $\hat{w}_0 = 0$ and the step-size $\alpha$ that satisfy $\alpha < \frac{1}{\kappa^2}$ [2] The progress made by gradient descent at each iteration can be slow and the idea behind acceleration is to use the information of the previous directions in order to improves the convergence rate of the algorithm.

**Heavy-ball**

Heavy-ball is a popular accelerated method that adds the *momentum* $\hat{w}_t - \hat{w}_{t-1}$ at each iteration

$$\hat{w}_{t+1} = \hat{w}_t - \alpha \, \mathrm{X}^* \left( \mathrm{X} \, \hat{w}_t - \mathbf{y} \right) + \beta(\hat{w}_t - \hat{w}_{t-1}) \tag{6}$$

with $\alpha, \beta \geq 0$, the case $\beta = 0$ reduces to gradient descent. In the quadratic case we consider it is also called Chebyshev iterative method. The optimization properties of heavy-ball have been studied extensively [24, 32]. Here, we consider the following variant. Let $\nu > 1$, consider the varying parameter heavy-ball replacing $\alpha, \beta$ in (6) with $\alpha_{t+1}, \beta_{t+1}$ defined as:

$$\alpha_t = \frac{4}{\kappa^2} \frac{(2t + 2\nu - 1)(t + \nu - 1)}{(t + 2\nu - 1)(2t + 4\nu - 1)} \qquad \beta_t = \frac{(t-1)(2t-3)(2t+2\nu-1)}{(t+2\nu-1)(2t+4\nu-1)(2t+2\nu-3)} \; ,$$

for $t > 0$ and with initialization $\hat{w}_{-1} = \hat{w}_0 = 0$, $\alpha_1 = \frac{1}{\kappa^2}\frac{4\nu+2}{4\nu+1}$, $\beta_1 = 0$. With this choice and considering the least-squares problem this algorithm is known as $\nu-$method in the inverse problem literature (see e.g. [11]). This seemingly complex parameters' choice allows to relates the approach to suitable orthogonal polynomials recursion as we discuss later.

**Nesterov acceleration**

The second form of gradient acceleration we consider is the popular Nesterov acceleration [22]. In our setting, it corresponds to the iteration

$$\hat{w}_{t+1} = \hat{v}_t - \alpha \, \mathrm{X}^* \left( \mathrm{X} \, \hat{v}_t - \mathbf{y} \right), \qquad \hat{v}_t = \hat{w}_t + \beta_t \left( \hat{w}_t - \hat{w}_{t-1} \right) \tag{7}$$

with the two initial points $\hat{w}_{-1} = \hat{w}_0 = 0$, and the sequence $\beta_t$ chosen as

$$\beta_t = \frac{t-1}{t+\beta} \; , \quad \beta \geq 1 \; . \tag{8}$$

Differently from heavy-ball, Nesterov acceleration uses the momentum term also in the evaluation of the gradient. Also in this case optimization results are well known [1, 29].

Here, as above, optimization results refer to solving ERM (3), (4), whereas in the following we study to which extent the above iterations can used to minimize the expected error (2). In the next section, we discuss a spectral approach which will be instrumental towards this goal.

## 3 Spectral filtering for accelerated methods

Least squares allows to consider spectral approaches to study the properties of gradient methods for learning. We illustrate these ideas for gradient descent before considering accelerated methods.

**Gradient descent as spectral filtering**

Note that by a simple (and classical) induction argument, gradient descent can be written as

$$\hat{w}_t = \alpha \sum_{j=0}^{t-1} (\mathrm{I} - \alpha\hat{\Sigma})^j \, \mathrm{X}^* \, \mathbf{y} \; .$$

Equivalently using spectral calculus

$$\hat{w}_t = g_t(\hat{\Sigma}) \, \mathrm{X}^* \, \mathbf{y} \; , \qquad \text{with} \quad \hat{\Sigma} = \mathrm{X}^* \, \mathrm{X},$$

where $g_t$ are the polynomials $g_t(\sigma) = \alpha \sum_{j=0}^{t-1} (\mathrm{I} - \alpha\sigma)^j$ for all $\sigma \in (0, \kappa^2]$ and $t \in \mathbb{N}$. Note that, the polynomials $g_t$ are bounded by $\alpha t$. A first observation is that $g_t(\sigma)\sigma$ converges to $1$ as $t \to \infty$, since $g_t(\sigma)$ converges to $\frac{1}{\sigma}$. A second observation is that the *residual polynomials* $r_t(\sigma) = 1 - \sigma g_t(\sigma)$, which are all bounded by $1$, control ERM convergence since,

$$\|\mathrm{X}\,\hat{w}_t - \mathbf{y}\|_n = \left\| \mathrm{X}\,g_t(\hat{\Sigma})\,\mathrm{X}^*\,\mathbf{y} - \mathbf{y} \right\|_n = \left\| g_t(\hat{\Sigma})\hat{\Sigma}\mathbf{y} - \mathbf{y} \right\|_n = \left\| r_t(\hat{\Sigma})\mathbf{y} \right\|_n \leq \left\| r_t(\hat{\Sigma}) \right\|_{op} \|\mathbf{y}\|_n \; .$$

In particular, if $\mathbf{y}$ is in the range of $\hat{\Sigma}^r$ for some $r > 0$ (source condition on $\mathbf{y}$) improved convergence rates can be derived noting that by an easy calculation

$$|r_t(\sigma)\sigma^q| \leq \left(\frac{q}{\alpha}\right)^q \left(\frac{1}{t}\right)^q \; .$$

As we show in Section 4, considering the polynomials $g_t$ and $r_t$ allows to study not only ERM but also expected risk minimization (2), by relating gradient methods to their infinite sample limit. Further, we show how similar reasoning hold for accelerated methods. In order to do so, it useful to first define the characterizing properties of $g_t$ and $r_t$.

## 3.1 Spectral filtering

The following definition abstracts the key properties of the function $g_t$ and $r_t$ often called *spectral filtering function* [2]. Following the classical definition we replace $t$ with a generic parameter $\lambda$.

**Definition 1.**
The family $\{g_\lambda\}_{\lambda \in (0,1]}$ is called *spectral filtering function* if the following conditions hold:

**(i)** There exist a constant $E < +\infty$ such that, for any $\lambda \in (0,1]$

$$\sup_{\sigma \in (0,\kappa^2]} |g_\lambda(\sigma)| \leq \frac{E}{\lambda} \ . \tag{9}$$

**(ii)** Let $r_\lambda(\sigma) = 1 - \sigma \, g_\lambda(\sigma)$ there exist a constant $F_0$ such that, for any $\lambda \in (0,1]$

$$\sup_{\sigma \in (0,\kappa^2]} |r_\lambda(\sigma)| \leq F_0 \ . \tag{10}$$

**Definition 2.** (Qualification)
 The qualification of the spectral filtering function $\{g_\lambda\}_\lambda$ is the maximum parameter $q$ such that for any $\lambda \in (0,1]$ there exist a constant $F_q$ such that

$$\sup_{\sigma \in (0,\kappa^2]} |r_\lambda(\sigma)\sigma^q| \leq F_q \lambda^q \ . \tag{11}$$

Moreover we say that a filtering function has qualification $\infty$ if (11) holds for every $q > 0$.

Methods with finite qualification might have slow convergence rates in certain regimes. The smallest the qualification the worse the rates can be.

The discussion in the previous section shows that gradient descent defines a spectral filtering function where $\lambda = 1/t$. More precisely, the following holds.

**Proposition 1.** *Assume $\lambda = \frac{1}{t}$ for $t \in \mathbb{N}$, then the polynomials $g_t$ related to the gradient descent iterates, defined in (5), are a filtering function with parameters $E = \alpha$ and $F_0 = 1$. Moreover it has qualification $\infty$ with parameters $F_q = (q/\alpha)^q$.*

The above result is classical and we report a proof in the appendix for completeness. Next, we discuss analogous results for accelerate methods and then compare the different spectral filtering functions.

## 3.2 Spectral filtering for accelerated methods

For the heavy-ball (6) the following result holds

**Proposition 2.** *Assume $\kappa \leq 1$, let $\nu > 0$ and $\lambda = \frac{1}{t^2}$ for $t \in \mathbb{N}$, then the polynomials $g_t$ related to heavy-ball method (6) are a filtering function with parameters $E = 2$ and $F_0 = 1$. Moreover there exist a positive constant $c_\nu < +\infty$ such that the $\nu$-method has qualification $\nu$.*

The proof of the above proposition follows combining several intermediate results from [11]. The key idea is to show that the residual polynomials defined by heavy-ball iteration form a sequence of orthogonal polynomials with respect to the weight function

$$\omega_\nu(\sigma) = \frac{\sigma^{2\nu}}{\sigma^{\frac{1}{2}} (1 - \sigma)^{\frac{1}{2}}} \ ,$$

which is a so called shifted Jacobi weight. Results from orthogonal polynomials can then be used to characterize the corresponding spectral filtering function.
The following proposition considers Nesterov acceleration.

**Proposition 3.** *Assume* $\lambda = 1/t^2$, *then the polynomials* $g_t$ *related to Nesterov iterates* (7) *are a filtering function with constants* $E = 2\alpha$ *and* $F_0 = 1$. *Moreover the qualification of this method is at least* $1/2$ *with constants* $F_q = \left(\frac{\beta^2}{\alpha}\right)^q$.

Filtering properties of the Nesterov iteration (7) have been studied recently in the context of inverse problems [23]. In the appendix 7.3 we provide a simplified proof based on studying the properties of suitable discrete dynamical systems defined by the Nesterov iteration (7).

### 3.3 Comparing the different filter functions

We summarize the properties of the spectral filtering function of the various methods for $\kappa = 1$.

| Method | E | $F_0$ | $F_q$ | Qualification |
|---|---|---|---|---|
| Gradien descent | 1 | 1 | $q^q$ | $\infty$ |
| Heavy-ball | 2 | 1 | $c_\nu \quad (q = \nu)$ | $\nu$ |
| Nesterov | 2 | 1 | $\beta^{2q}$ | $\geq 1/2$ |

The main observation is that the properties of the spectral filtering functions corresponding to the different iterations depend on $\lambda = 1/t$ for gradient descent, but on $\lambda = 1/t^2$ for the accelerated methods. As we see in the next section this leads to substantially different learning properties. Further we can see that gradient descent is the only algorithm with qualification $\infty$, even if the parameter $F_q = q^q$ can be very large. The accelerated methods seem to have smaller qualification. In particular, the heavy-ball method can attain a high qualification, depending on $\nu$, but the constant $c_\nu$ is unknown and could be large. For Nesterov accelerated method, the qualification is at least $1/2$ and it's an open question whether this bound is tight or higher qualification can be attained.

In the next section, we show how the properties of the spectral filtering functions can be exploited to study the excess risk of the corresponding iterations.

## 4 Learning properties for accelerated methods

We first consider a basic scenario and then a more refined analysis leading to a more general setting and potentially faster learning rates.

### 4.1 Attainable case

Consider the following basic assumption.

**Assumption 1.** *Assume there exist* $M > 0$ *such that* $|y| < M$ $\rho$-*almost surely and* $w^* \in \mathcal{X}$ *such that* $\mathcal{E}(w^*) = \inf_{\mathcal{X}} \mathcal{E}$.

Then the following result can be derived.

**Theorem 1.** *Under Assumption 1, let* $\hat{w}_t^{GD}$ *and* $\hat{w}_t^{acc}$ *be the* $t$-*th iterations respectively of gradient descent* (5) *and an accelerated version given by* (6) *or* (7). *Assuming the sample-size* $n$ *to be large enough and let* $\delta \in (0, 1/2)$ *then there exist two positive constant* $C_1$ *and* $C_2$ *such that with probability at least* $1 - \delta$

$$\mathcal{E}(\hat{w}_t^{GD}) - \inf_{\mathcal{H}} \mathcal{E} \leq C_1 \left(\frac{1}{t} + \frac{t}{n}\right) \log^2 \frac{2}{\delta}$$

$$\mathcal{E}(\hat{w}_t^{acc}) - \inf_{\mathcal{H}} \mathcal{E} \leq C_2 \left(\frac{1}{t^2} + \frac{t^2}{n}\right) \log^2 \frac{2}{\delta} .$$

*where the constants* $C_1$ *and* $C_2$ *do not depend on* $n, t, \delta$, *but depend on the chosen optimization method.*
*Moreover by choosing the stopping rules* $t^{GD} = O(n^{1/2})$ *and* $t^{acc} = O(n^{1/4})$ *both algorithms have learning rate of order* $\frac{1}{\sqrt{n}}$.

The proof of the above results is given in the appendix and the novel part is the one concerning accelerated methods, particularly Nesterov acceleration. The result shows how the number of iteration

controls the learning properties both for gradient descent and accelerated gradient. In this sense implicit regularization occurs in all these approaches. For any $t$ the error is split in two contributions. Inspecting the proof it is easy to see that, the first term in the bound comes from the convergence properties of the algorithm with infinite data. Hence the optimization error translates into a bias term. The decay for accelerated method is much faster than for gradient descent. The second term arises from comparing the empirical iterates with their infinite sample (population) limit. It is a variance term depending on the sampling in the data and hence decreases with the sample size. For all methods, this term increases with the number of iterations, indicating that the empirical and population iterations are increasingly different. However, the behavior is markedly worse for accelerated methods. The benefit of acceleration seems to be balanced out by this more unstable behavior. In fact, the benefit of acceleration is apparent balancing the error terms to obtain a final bound. The obtained bound is the same for gradient descent and accelerated methods, and is indeed optimal since it matches corresponding lower bounds [3, 7]. However, the number of iterations needed by accelerated methods is the square root of those needed by gradient descent, indicating a substantial computational gain can be attained. Next we show how these results can be generalized to a more general setting, considering both weaker and stronger assumptions, corresponding to harder or easier learning problems.

## 4.2 More refined result

Theorem 1 is a simplified version of the more general result that we discuss in this section. We are interested in covering also the non-attainable case, that is when there is no $w^* \in \mathcal{X}$ such that $\mathcal{E}(w^*) = \inf_{\mathcal{X}} \mathcal{E}$. In order to cover this case we have to introduce several more definitions and notations. In Appendix 8.2 we give a more detailed description of the general setting. Consider the space $L^2_{\rho_{\mathcal{X}}}$ of the square integrable functions with the norm $\|f\|^2_{\rho_{\mathcal{X}}} = \int_{\mathcal{X}} f(x)^2 \, d\rho_{\mathcal{X}}(x)$ and extend the expected risk to $L^2_{\rho_{\mathcal{X}}}$ defining $\mathcal{E}(f) = \int_{\mathcal{X} \times \mathbb{R}} (f(x) - y)^2 \, d\rho(x, y)$. Let $\mathcal{H} \subseteq L^2_{\rho_{\mathcal{X}}}$ be the hypothesis space of functions such that $f(x) = \langle w, x \rangle \, \rho_{\mathcal{X}}$ almost surely. Recall that, the minimizer of the expected risk over $L^2_{\rho_{\mathcal{X}}}$ is the regression function $f_\rho = \int_{\mathcal{X}} y \, d\rho(y|x)$. The projection $f_{\mathcal{H}}$ over the closure of the hypothesis space $\mathcal{H}$ is defined as

$$f_{\mathcal{H}} = \arg \min_{g \in \overline{\mathcal{H}}} \|g - f_\rho\|_{\rho_{\mathcal{X}}} \ .$$

Let $L : L^2_{\rho_{\mathcal{X}}} \to L^2_{\rho_{\mathcal{X}}}$ be the integral operator

$$Lf(x) = \int_{\mathcal{X}} f(x') \langle x, x' \rangle \, d\rho_{\mathcal{X}}(x') \ .$$

The first assumption we consider concern the moments of the output variable and is more general than assuming the output variable $y$ to be bounded as assumed before.

**Assumption 2.** *There exist positive constant $Q$ and $M$ such that for all $\mathbb{N} \ni l \geq 2$,*

$$\int_{\mathbb{R}} |y|^l \, d\rho(y|x) \leq \frac{1}{2} l! M^{l-2} Q^2 \qquad \rho_{\mathcal{X}} \text{ almost surely.}$$

This assumption is standard and satisfied in classification or regression with well behaved noise. Under this assumption the regression function $f_\rho$ is bounded almost surely

$$|f_\rho(x)| \leq \int_{\mathbb{R}} |y| \, d\rho(y|x) \leq \left( \int_{\mathbb{R}} |y|^2 \, d\rho(y|x) \right)^{1/2} \leq Q \ . \tag{12}$$

The next assumptions are related to the regularity of the target function $f_{\mathcal{H}}$.

**Assumption 3.**
*There exist a positive constant $B$ such that the target function $f_{\mathcal{H}}$ satisfy*

$$\int_{\mathcal{X}} (f_{\mathcal{H}}(x) - f_\rho(x))^2 \, x \otimes x \, d\rho_{\mathcal{X}}(x) \preceq B^2 \Sigma \ .$$

This assumption is needed to deal with the misspecification of the model. The last assumptions quantify the regularity of $f_{\mathcal{H}}$ and the size (capacity) of the space $\mathcal{H}$.

**Assumption 4.**
*There exist $g_0 \in L^2_{\rho_X}$ and $r > 0$ such that*

$$f_{\mathcal{H}} = L^r g_0\,, \quad with \; \|g_0\|_{\rho_X} \leq R.$$

*Moreover we assume that there exist $\gamma \geq 1$ and a positive constant $c_\gamma$ such that the effective dimension*

$$\mathcal{N}(\lambda) = \mathrm{Tr}\left(L\,(L + \lambda \mathrm{I})^{-1}\right) \leq c_\gamma \lambda^{-\frac{1}{\gamma}}\,.$$

The assumption on $\mathcal{N}(\lambda)$ is always true for $\gamma = 1$ and $c_1 = \kappa^2$ and it's satisfied when the eigenvalues $\sigma_i$ of $L$ decay as $i^{-\gamma}$. We recall that, the space $\mathcal{H}$ can be characterized in terms of the operator $L$, indeed

$$\mathcal{H} = L^{1/2}\left(L^2_{\rho_X}\right).$$

Hence, the non-attainable corresponds to considering $r < 1/2$.

**Theorem 2.** *Under Assumption 2, 3, 4, let $\hat{w}^{GD}_t$ and $\hat{w}^{acc}_t$ be the $t$-th iterations of gradient descent (5) and an accelerated version given by (6) or (7) respectively. Assuming the sample-size $n$ to be large enough, let $\delta \in (0, 1/2)$ and assuming $r$ to be smaller than the qualification of the considered algorithm (and equal to $1/2$ in the case of Nesterov accelerated methods), then there exist two positive constant $C_1$ and $C_2$ such that with probability at least $1 - \delta$*

$$\mathcal{E}(\hat{w}^{GD}_t) - \inf_{\mathcal{H}} \mathcal{E} \leq C_1 \left(\frac{1}{t^{2r}} + \frac{t^{\frac{1}{\gamma}}}{n}\right) \log^2 \frac{2}{\delta}$$

$$\mathcal{E}(\hat{w}^{acc}_t) - \inf_{\mathcal{H}} \mathcal{E} \leq C_2 \left(\frac{1}{t^{4r}} + \frac{t^{\frac{2}{\gamma}}}{n}\right) \log^2 \frac{2}{\delta}\,.$$

*where the constants $C_1$ and $C_2$ do not depend on $n, t, \delta$, but depend on the chosen optimization. Choosing the stopping rules $t^{GD} = O(n^{\frac{\gamma}{2\gamma r + 1}})$ and $t^{acc} = O(n^{\frac{\gamma}{4\gamma r + 2}})$ both gradient descent and accelerated methods achieve a learning rate of order $O\left(n^{\frac{-2\gamma r}{2\gamma r + 1}}\right)$.*

The only reason why we do not consider $r < 1/2$ in the analysis of Nesterov accelerated methods is that our proof require the qualification of the method to be larger than 1 for technical reasons. However we think that our result can be extended to that case, furthermore we think Nesterov qualification to be larger than 1, however it's an open question whether higher qualification can be attained. The proof of the above result is given in the appendix. The general structure of the bound is the same as in the basic setting, which is now recovered as a special case. However, in this more general form, the various terms in the bound depend now on the regularity assumptions on the problem. In particular, the variance depends on the effective dimension behavior, e.g. on the eigenvalue decay, while the bias depend on the regularity assumption on $f_{\mathcal{H}}$. The general comparison between gradient descent and accelerated methods follows the same line as in the previous section. Faster bias decay of accelerated methods is contrasted by a more unstable behavior. As before, the benefit of accelerated methods becomes clear when deriving optimal stopping time and corresponding learning bound: they achieve the accuracy of gradient methods but in considerable less time. While heavy-ball and Nesterov have again similar behaviors, here a subtle difference resides in their different qualifications, which in principle lead to different behavior for easy problems, that is for large $r$ and $\gamma$. In this regime, gradient descent could work better since it has infinite qualification. For problems in which $r < 1/2$ and $\gamma = 1$ the rates are worse than in the basic setting, hence these problems are hard.

## 4.3 Related work

In the convex optimization framework a similar phenomenon was pointed out in [10] where they introduce the notion of inexact first-order oracle and study the behaviour of several first-order methods of smooth convex optimization with such oracle. In particular they show that the superiority of accelerated methods over standard gradient descent is no longer absolute when an inexact oracle is used. This because acceleration suffer from the accumulation of the errors committed by the inexact oracle. A relevant result on the generalization properties of learning algorithm is [5] in which they introduce the notion of *uniform* stability and use it to obtain generalization error bounds

for regularization based learning algorithms. Recently, to show the effectiveness of commonly used optimization algorithms in many large-scale learning problems, algorithmic stability has been established for stochastic gradient methods [14], as well as for any algorithm in situations where global minima are approximately achieved [8]. For Nesterov's accelerated gradient descent and heavy-ball method, [9] provide stability upper bounds for quadratic loss function in a finite dimensional setting. All these approaches focus on the definition of uniform stability given in [5]. Our approach to the stability of a learning algorithm is based on the study of filtering properties of accelerated methods together with concentration inequalities, we obtain upper bounds on the generalization error for quadratic loss in a infinite dimensional Hilbert space and generalize the bounds obtained in [9] by considering different regularity assumptions and by relaxing the hypothesis of the existence of a minimizer of the expected risk on the hypothesis space.

## 5    Numerical simulation

In this section we show some numerical simulations to validate our results. We want to simulate the case in which the eigenvalues $\sigma_i$ of the operator $L$ are $\sigma_i = i^{-\gamma}$ for some $\gamma \leq 1$ and the non-attainable case $r < 1/2$. In order to do this we observe that if we consider the kernel setting over a finite space $\mathcal{Z} = \{z_1, \ldots, z_n\}$ of size $N$ with the uniform probability distribution $\rho_{\mathcal{Z}}$, then the space $L^2(\mathcal{Z}, \rho_{\mathcal{Z}})$ becomes $\mathbb{R}^N$ with the usual scalar product multiplied by $1/N$. the operator $L$ becomes a $N \times N$ matrix which entries are $L_{i,j} = K(z_i, z_j)$ for every $i, j \in \{1, \ldots, N\}$, where $K$ is the kernel, which is fixed by the choice of the matrix $L$. We build the matrix $L = UDU^T$ with $U \in \mathbb{R}^{N \times N}$ orthogonal matrix and $D$ diagonal matrix with entries $D_{i,i} = i^{-\gamma}$. The source condition becomes $f_{\mathcal{H}} = L^r g_0$ for some $g_0 \in \mathbb{R}^N, r > 0$. We simulate the observed output as $y = f_{\mathcal{H}} + \mathcal{N}(0, \sigma)$ where $\mathcal{N}(0, \sigma)$ is the standardx normal distribution of variance $\sigma^2$. The sampling operation can be seen as extracting $n$ indices $i_1, \ldots, i_n$ and building the kernel matrix $\hat{K}_{j,k} = K(z_{i_j}, z_{i_k})$ and the noisy labels $\hat{y}_j = y_{i_j}$ for every $j, k \in \{1, \ldots, n\}$. The Representer Theorem ensure that we can built our estimator $\hat{f} \in \mathbb{R}^N$ as $\hat{f}(z) = \sum_{j=1}^{n} K(z, z_{i_j}) c_j$ where the vector $c$ depends on the chosen optimization algorithm and takes the form $c = g_t(\hat{K})y$. The excess risk of the estimator $\hat{f}$ is given by $\|\hat{f} - f_{\mathcal{H}}\|^2_{L^2_{\mathcal{Z}}}$.

For every algorithm considered, we run 50 repetitions, in which we sample the data-space and compute the error $\|\hat{f}_t - f_{\mathcal{H}}\|^2_{L^2_{\mathcal{Z}}}$, where $\hat{f}_t$ represents the estimator related to the $t$-th iteration of one of the considered algorithms, and in the end we compute the mean and the variance of those errors. In Figure 1 we simulate the error of all the algorithms considered for both attainable and non-attainable case. We observe that both heavy-ball and Nesterov acceleration provides faster convergence rates with respect to gradient descent method, but the learning accuracy is not improved. We observe that the accelerated methods considered show similar behavior and that for "easy problem" (large $r$) that gradient descent can exploits its higher qualification and perform similarly to the accelerated methods.

In Figure 2 we show the test error related to the real dataset *pumadyn8nh* (available at https://www.dcc.fc.up.pt/ ltorgo/Regression/puma.html). Even in this case we can observe the behaviors shown in our theoretical results.

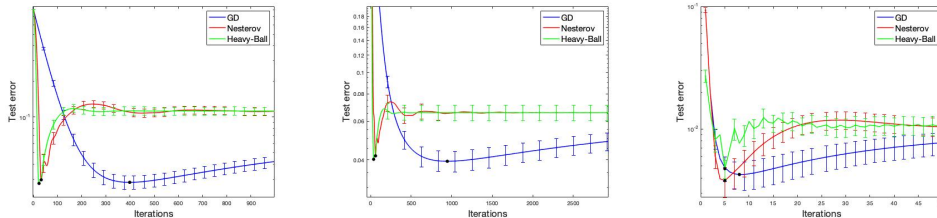

Fig. 1: Mean and variance of error $\|\hat{f}_t - f_{\mathcal{H}}\|^2_N$ for the $t$-th iteration of gradient descent (GD), Nesterov accelerated algorithm and heavy-ball ($\nu = 1$). Black dots shows the absolute minimum of the curves. The parameters are chosen $N = 10^4, n = 10^2, \gamma = 1, \sigma = 0.5$. We show the attainable case ($r = 1/2$) in the left, the "hard case" ($r = 0.1 < 1/2$) in the center and the "easy case" (r=2>1/2) in the right.

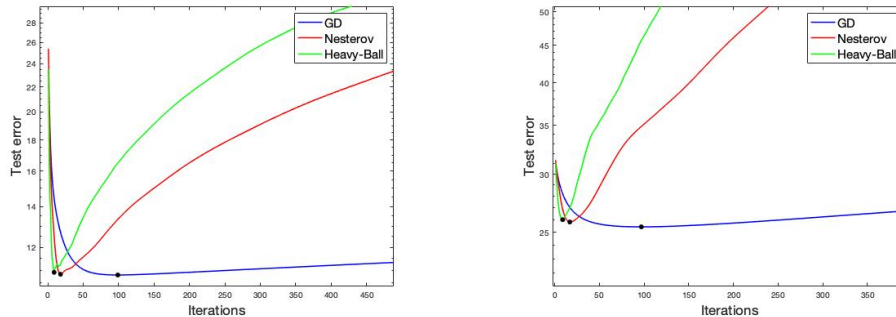

Fig. 2: Test error on the real dataset *pumadyn8nh* using gradient descent (GD), Nesterov accelerated algorithm and heavy-ball. In the left we use a gaussian kernel with $\sigma = 1.2$ and in the right a polynomial kernel of degree 9.

# 6 Conclusion

In this paper, we have considered the implicit regularization properties of accelerated gradient methods for least squares in Hilbert space. Using spectral calculus we have characterized the properties of the different iterations in terms of suitable polynomials. Using the latter, we have derived error bounds in terms of suitable bias and variance terms. The main conclusion is that under the considered assumptions accelerated methods have smaller bias but also larger variance. As a byproduct they achieve the same accuracy of vanilla gradient descent but with much fewer iterations. Our study opens a number of potential theoretical and empirical research directions. From a theory point of view, it would be interesting to consider other learning regimes, for examples classification problems, different loss functions or other regularity assumptions beyond classical nonparametric assumptions, e.g. misspecified models and fast eigenvalues decays (Gaussian kernel). From an empirical point of view it would be interesting to do a more thorough investigation on a larger number of simulated and real data-sets of varying dimension.

## Acknowledgments

This material is based upon work supported by the Center for Brains, Minds and Machines (CBMM), funded by NSF STC award CCF-1231216, and the Italian Institute of Technology. We gratefully acknowledge the support of NVIDIA Corporation for the donation of the Titan Xp GPUs and the Tesla k40 GPU used for this research. L. R. acknowledges the financial support of the AFOSR projects FA9550-17-1-0390 and BAA-AFRL-AFOSR-2016-0007 (European Office of Aerospace Research and Development), and the EU H2020-MSCA-RISE project NoMADS - DLV-777826. N.P. would like to thank Murata Tomoya for the useful observations.

## Footnotes

[1] As shown in Appendix this choice allows to recover nonparametric kernel learning as a special case.

[2] The step-size $\alpha$ is the step-size at the $t$-th iteration and satisfies the condition $0 < \alpha \|\mathrm{X}\|_{op}^2 < 1$ , where $\|\cdot\|_{op}$ denotes the operatorial norm. Since the operator X is bounded by $\kappa$ (which means $\|\mathrm{X}\|_{op} \leq \kappa$) it is sufficient to assume $\alpha < \frac{1}{\kappa^2}$.

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
