[Supplementary Material]

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

# 7   Appendix: regularization properties for accelerated algorithms

## 7.1   Regularization properties for gradient descent

*Proof of Proposition 1*

*Proof.*
Since $\sigma \in (0, \kappa^2]$ and $\alpha$ is chosen such that $\alpha \leq \frac{1}{\kappa^2}$ it holds that $(1 - \alpha\sigma) \leq 1$ for every and so for the definitions of $g_t$ and $r_t$ it holds

$$\sup_{\sigma \in (0, \kappa^2]} |g_t(\sigma)| \leq \alpha t$$

$$\sup_{\sigma \in (0, \kappa^2]} |r_t(\sigma)| \leq 1$$

hence Landweber polynomials verify (9) and (10) with $E = \alpha$ and $F_0 = 1$.
For what concern the qualification of this method, for every $q \geq 0$ the maximum of the function $r_t(\sigma)\sigma^q$ is attained at $\sigma = \frac{1}{\alpha}\frac{q}{t+q}$, so we get

$$0 \leq r_t(\sigma)\sigma^q \leq \left(\frac{1}{\alpha}\right)^q \left(\frac{q}{t+q}\right)^q \leq \left(\frac{q}{\alpha}\right)^q \left(\frac{1}{t}\right)^q \ ,$$

hence we prove (11) for every $q \geq 0$ with $F_q = \left(\frac{q}{\alpha}\right)^q$ and complete the proof.

$\square$

## 7.2   Regularization properties for heavy-ball

For the sake of simplicity assume $\kappa \leq 1$. Before proceeding with the analysis of the $\nu$-method we state one lemma, which will be useful in the following.

**Lemma 1.**
*Let $g_t$ be a family of polynomials of degree $t - 1$ with $t \in \mathbb{N}$ and $r_t$ the associated residuals. Assume the residuals satisfy*

$$|r_t(\sigma)| \leq 1 \qquad \forall \sigma \in [0,1],\ t \in \mathbb{N} \tag{13}$$

*then it holds that*

$$|g_t(\sigma)| \leq 2t^2 \qquad \forall \sigma \in [0,1] \ .$$

*Proof.*
Using the definition of the residual and the Mean Value Theorem there exist $\bar{\sigma} \in [0, \sigma]$ such that

$$g_t(\sigma) = \frac{1 - r_t(\sigma)}{\sigma} = \frac{r_t(0) - r_t(\sigma)}{\sigma} = -r_t'(\bar{\sigma}) \ .$$

where $r_t'$ denotes the first derivative of $r_t$.
Markov's inequality for polynomials implies that

$$\sup_{\sigma \in [0,1]} |r_t'(\sigma)| \leq 2t^2 \sup_{\sigma \in [0,1]} |r_t(\sigma)| \ ,$$

hence it holds

$$g_t(\sigma) \leq \sup_{\bar{\sigma} \in [0,1]} |r_t'(\bar{\sigma})| \leq 2t^2 \ .$$

$\square$

Fixed $\nu > 0$ the residual polynomials $\{r_t\}_t$ associated to the $\nu$-method form a sequence of orthogonal polynomials with respect to the weight function

$$\omega_\nu(\sigma) = \frac{\sigma^{2\nu}}{\sigma^{\frac{1}{2}} (1 - \sigma)^{\frac{1}{2}}} \ ,$$

which is a shifted Jacobi weight, hence the residual polynomials $\{r_t\}_t$ are normalized shifted copies of Jacobi polynomials, where the normalization is due to the constraint $r_t(0) = 1$.

Thanks to the properties of orthogonal polynomials, they satisfy Christoffel-Darboux recurrence formula (see e.g. [30])

$$r_{t+1} = r_t(\sigma) + \beta_{t+1}\left(r_t(\sigma) - r_{t-1}(\sigma)\right) - \alpha_{t+1}\sigma r_t(\sigma) , \qquad t \geq 1$$

and a straightforward computation shows that this recursion on our problem carries over to the iterates $\hat{w}_t$ of the associated method

$$\hat{w}_{t+1} = \hat{w}_t - \alpha_{t+1} \, \mathrm{X}^* \left(\mathrm{X}\, \hat{w}_t - \mathbf{y}\right) + \beta_{t+1}(\hat{w}_t - \hat{w}_{t-1}) .$$

where, for every $t > 1$, the parameters $\alpha_t$, $\beta_t$ are defined by

$$\alpha_t = 4\frac{(2t + 2\nu - 1)(t + \nu - 1)}{(t + 2\nu - 1)(2t + 4\nu - 1)}$$

$$\beta_t = \frac{(t-1)(2t-3)(2t + 2\nu - 1)}{(t + 2\nu - 1)(2t + 4\nu - 1)(2t + 2\nu - 3)} ,$$

with initialization $\hat{w}_{-1} = \hat{w}_0 = 0$, $\alpha_1 = \frac{4\nu+2}{4\nu+1}$, $\beta_1 = 0$.
In particular it holds the following result from [11].

**Theorem 3.**
*The residual polynomials $\{r_t\}_t$ of the $\nu$-method ($\nu$ fixed) are uniformly bounded for all $t \in \mathbb{N}$,*

$$|r_t(\sigma)| \leq 1 \qquad \sigma \in [0,1];$$

*they further satisfy*

$$|\sigma^\nu r_t(\sigma)| \leq c_\nu t^{-2\nu} \qquad \forall \sigma \in [0,1] \tag{14}$$

*with appropriate constants $0 < c_\nu < +\infty$.*

## *Proof of Proposition 2*

*Proof.*
Theorem 3 states that (10) holds true with $F_0 = 1$ and that the qualification of the method is $\nu$. Moreover by the Lemma 1 we get that also that (9) holds with $E = 1$.

$\square$

### 7.3 Regularization properties for Nesterov's acceleration

Nesterov iterates (7) can be written as

$$\hat{w}_{t+1} = \hat{w}_t + \beta_t \left(\hat{w}_t - \hat{w}_{t-1}\right) - \alpha\, \mathrm{X}^* \left(\mathrm{X}\left(\hat{w}_t + \beta_t\left(\hat{w}_t - \hat{w}_{t-1}\right)\right) - \mathbf{y}\right) =$$

$$= \left[(\beta_t + 1)\left(\mathrm{I} - \alpha\hat{\Sigma}\right)\right]\hat{w}_t + \left[-\beta_t\left(\mathrm{I} - \alpha\hat{\Sigma}\right)\right]\hat{w}_{t-1} + \alpha\, \mathrm{X}^* \mathbf{y}$$

and since $\hat{w}_t = g_t\left(\hat{\Sigma}\right)\mathrm{X}^*\mathbf{y}$ it can be easily proved that the polynomials $g_t$ and the residual $r_t$ satisfy the following recursions

$$g_{t+1}(\sigma) = (1 - \alpha\sigma)\left[g_t(\sigma) + \beta_t\left(g_t(\sigma) - g_{t-1}(\sigma)\right)\right] + \alpha \tag{15}$$

$$r_{t+1}(\sigma) = (1 - \alpha\sigma)\left[r_t(\sigma) + \beta_t\left(r_t(\sigma) - r_{t-1}(\sigma)\right)\right] \tag{16}$$

for every $t \in \mathbb{N}$ with initialization $g_{-1} = g_0 = 0$ and $r_{-1} = r_0 = 1$.
Moreover, we can rewrite (16) as

$$r_{t+1}(\sigma) = (1 - \alpha\sigma)\left[(1 - \theta_t)r_t(\sigma) + \theta_t\left(r_{t-1}(\sigma) + \frac{1}{\theta_{t-1}}(r_t(\sigma) - r_{t-1}(\sigma))\right)\right] \tag{17}$$

where for every $t \in \mathbb{N}$ $\theta_t$ is defined such that

$$\beta_t = \frac{\theta_t}{\theta_{t-1}}(1 - \theta_{t-1}) ,$$

in particular, the choice (8) implies

$$\theta_t = \frac{\beta}{t + \beta} . \tag{18}$$

With these choices we can state a first proposition about the properties of the residual polynomials of the Nesterov's accelerated method.

**Proposition 4.**
*Let $r_t$ satisfy the recursion* (17) *where the step-size $\alpha$ is chosen such that $\alpha\kappa^2 < 1$ and $\theta_t$ defined in* (18)*, the for all $r \in [0, 1/2]$*

$$\sigma^r |r_t(\sigma)| \leq \left(\frac{\beta^2}{\alpha}\right)^r t^{-2r} \tag{19}$$

*for all $\sigma \in [0, \kappa^2]$.*

*Proof.*
Let $\sigma \in [0, \kappa^2]$, following [23] we can see the right hand of (17) as a convex combination between $r_t$ and

$$R_t(\sigma) = r_{t-1}(\sigma) + \frac{1}{\theta_{t-1}}(r_t(\sigma) - r_{t-1}(\sigma)) .$$

We can observe that polynomials $r_t$ and $R_t$ satisfy the following recursions

$$r_{t+1}(\sigma) = (1 - \alpha\sigma)(1 - \theta_t)r_t(\sigma) + \theta_t(1 - \alpha\sigma)R_t(\sigma)$$

$$R_{t+1}(\sigma) = -\frac{\alpha\sigma}{\theta_t}(1 - \theta_t)r_t(\sigma) + (1 - \alpha\sigma)R_t(\sigma)$$

By computing the square of the polynomials and rescaling them in order to get the two mixed term to be opposite, we obtain that

$$\frac{\alpha\sigma}{\theta_t^2}r_{t+1}^2(\sigma) + (1 - \alpha\sigma)R_{t+1}^2 = (1 - \alpha\sigma)\left[\frac{(1 - \theta_t)^2\alpha\sigma}{\theta_t^2}r_t^2(\sigma) + (1 - \alpha\sigma)R_t^2(\sigma)\right]$$

We can observe that parameters $\theta_t$ satisfy the following

$$1 \geq \theta_t \geq \frac{\theta_{t-1}}{1 + \theta_{t-1}}$$

which implies

$$\frac{(1 - \theta_t^2)}{\theta_t^2} \leq \frac{1}{\theta_{t-1}^2} .$$

Hence we get that

$$\frac{\alpha\sigma}{\theta_t^2}r_{t+1}^2(\sigma) + (1 - \alpha\sigma)R_{t+1}^2 \leq (1 - \alpha\sigma)\left[\frac{\alpha\sigma}{\theta_{t-1}^2}r_t^2(\sigma) + (1 - \alpha\sigma)R_t^2(\sigma)\right]$$

$$\leq (1 - \alpha\sigma)^t\left[\frac{\alpha\sigma}{\theta_0^2}r_1^2(\sigma) + (1 - \alpha\sigma)R_1^2(\sigma)\right]$$

where the second inequality follows by induction.
Finally, using that $\theta_0 = 1$ and $R_0 = 1$, yields that

$$\frac{\alpha\sigma}{\theta_{t-1}^2}r_t^2(\sigma) + (1 - \alpha\sigma)R_t^2 \leq (1 - \alpha\sigma)^{t+1} .$$

This inequality implies that both the terms in the sum are smaller that $(1 - \alpha\sigma)^{t+1}$, hence

$$|R_t(\sigma)| \leq 1 \tag{20}$$

$$\sigma r_t^2(\sigma) \leq \frac{\theta_{t-1}^2}{\alpha}(1 - \alpha\sigma)^{t+1} \tag{21}$$

By induction it follows from (20) that (19) holds for $r = 0$:

$$|r_t(\sigma)| \leq 1$$

because $r_{t+1}$ is a convex combination of $r_t$ and $R_t$ multiplied by $(1 - \alpha\sigma)$.
While (21) implies (19) for $r = 1/2$. The remaining cases $r \in (0, 1/2)$ follow by interpolation.
$\square$

By a scaled version of Lemma 1 it holds that

$$|g_t(\sigma)| \leq 2\alpha t^2 \qquad \forall \sigma \in [0, \kappa^2] .$$

*Proof of Proposition 3*

*Proof.* The proof follows immediately by the above results. $\square$

# 8 Appendix: generalization bound via spectral/regularized algorithm

## 8.1 Learning with kernels

The setting in this paper recover non-parametric regression over a RKHS as a special case. Let $\Xi \times \mathbb{R}$ be a probability space with distribution $\mu$, the goal is to minimize the risk

$$\mathcal{E}(f) = \int_{\Xi \times \mathbb{R}} (y - f(\xi))^2 \, d\mu(\xi, y).$$

A common way to build an estimator is to consider a symmetric kernel $K : \Xi \times \Xi \to \mathbb{R}$ which is positive definite, which means that for every $m \in \mathbb{N}$ and $\xi_1, \ldots, \xi_n \in \Xi$ the matrix with the entries $K(\xi_i, \xi_j)$ for $i, j = 1, \ldots, m$. This kernel defines a unique Hilbert space of function $\mathcal{H}_K$ with the inner product $\langle \cdot, \cdot \rangle_K$ and such that for all $\xi \in \Xi$, $K_\xi(\cdot) = K(\xi, \cdot) \in \mathcal{H}_K$ and the following reproducing property holds for all $f \in \mathcal{H}_K$, $f(\xi) = \langle f, K_\xi \rangle_K$. By introducing the feature map $\Psi : \Xi \to \mathcal{H}_K$ defined by $\Psi(\xi) = K_\xi$, and we further consider $\overline{\Psi} : \Xi \times \mathbb{R} \to \mathcal{H}_K \times \mathbb{R}$, where $\overline{\Psi}(\xi, y) = (K_\xi, y)$, which provide $\mathcal{H}_K \times \mathbb{R}$ the probability distribution $\mu_{\overline{\Psi}}$. Denoting $\mathcal{X} = \mathcal{H}_K$ and $\rho = \mu_{\overline{\Psi}}$ we come back to our previous setting, in fact by the change of variable $(K_\xi, y) = (x, y)$ we have

$$\int_{\Xi \times \mathbb{R}} (y - f(\xi))^2 \, d\mu(\xi, y) = \int_{\Xi \times \mathbb{R}} (y - \langle f, K_\xi \rangle_K)^2 \, d\mu(\xi, y) = \int_{\mathcal{X} \times \mathbb{R}} (y - \langle f, x \rangle_K)^2 \, d\rho(x, y) \ .$$

## 8.2 Mathematical setting

Let's consider the hypothesis space

$$\mathcal{H} = \left\{ f : \mathcal{X} \to \mathbb{R} \mid \exists \, w \in \mathcal{X} \text{ with } f(x) = \langle w, x \rangle \ \rho_{\mathcal{X}}\text{-almost surely} \right\},$$

which under assumptio 1 is a subspace of the Hilbert space of the square integral functions from $\mathcal{X}$ to $\mathbb{R}$ with respect to the measure $\rho_{\mathcal{X}}$

$$L_{\rho_{\mathcal{X}}}^2 = \left\{ f : \mathcal{X} \to \mathbb{R} \mid \|f\|_{\rho_{\mathcal{X}}}^2 = \langle f, f \rangle_{\rho_{\mathcal{X}}} := \int_{\mathcal{X}} f(x)^2 \, d\rho_{\mathcal{X}}(x) < +\infty \right\} \ .$$

The function that minimizes the expected risk over all possible measurable functions is the regression function [28].

$$f_\rho = \arg\min_{f : \mathcal{X} \to \mathbb{R}} \mathcal{E}(f), \quad \mathcal{E}(f) = \int_{\mathcal{X} \times \mathbb{R}} (f(x) - y)^2 \, d\rho(x, y)$$

$$f_\rho(x) = \int_{\mathbb{R}} y \, d\rho(y|x) \qquad \forall x \in \mathcal{X}, \ \rho_{\mathcal{X}}\text{-almost surely.}$$

which under assumption 1 the regression function $f_\rho$ belongs to $L_{\rho_{\mathcal{X}}}^2$.
Assuming (1) implies that a solution $f_{\mathcal{H}}$ for the problem

$$\inf_{\mathcal{H}} \mathcal{E},$$

which is equivalent to 2, is the projection of the regression function $f_\rho$ into the closure of $\mathcal{H}$ in $L_{\rho_{\mathcal{X}}}^2$. In fact a standard result (see e.g. [28]) show that for all $f$ in $L_{\rho_{\mathcal{X}}}^2$

$$\mathcal{E}(f) = \|f - f_\rho\|_{\rho_{\mathcal{X}}}^2 + \mathcal{E}(f_\rho) \ . \tag{22}$$

We now introduce some useful operators. Let $S : \mathcal{X} \to L_{\rho_{\mathcal{X}}}^2$ be the linear map defined by $Sw = \langle w, \cdot \rangle$ which is bounded by $k$ for (1), in fact for the Cauchy–Schwarz inequality

$$\|Sw\|_{\rho_{\mathcal{X}}}^2 = \int_{\mathcal{X}} \langle w, x \rangle^2 \, d\rho_{\mathcal{X}}(x) \leq \kappa^2 \|w\|^2 \ .$$

Furthermore, we consider the the adjoint operator $S^* : L_{\rho_{\mathcal{X}}}^2 \to \mathcal{X}$ (i.e. the operator which satisfy $\langle Sw, f \rangle_{\rho_{\mathcal{X}}} = \langle w, S^* f \rangle$), the covariance operator $\Sigma : \mathcal{X} \to \mathcal{X}$ given by $\Sigma = S^* S$ and the operator $L : L_{\rho_{\mathcal{X}}}^2 \to L_{\rho_{\mathcal{X}}}^2$ defined by $L = SS^*$. It's easy to observe that these operators are defined as follows

$$S^* f = \int_{\mathcal{X}} x f(x) \, d\rho_{\mathcal{X}}(x), \qquad \Sigma w = \int_{\mathcal{X}} \langle w, x \rangle x \, d\rho_{\mathcal{X}}(x), \qquad Lf = \int_{\mathcal{X}} f(x) \langle x, \cdot \rangle \, d\rho_{\mathcal{X}}(x)$$

and that the operators $\Sigma$ and $L$ are linear positive-definite trace class operators bounded by $\kappa^2$. Moreover, for any $w \in \mathfrak{X}$ it holds the following isometry property [28]

$$\|Sw\|_{\rho_{\mathfrak{X}}} = \left\|\sqrt{\Sigma}w\right\| .$$

Similarly we define the sampling operator $\mathrm{X} : \mathfrak{X} \to \mathbb{R}^n$ by $(\mathrm{X}\,w)_i = \langle w, x_i \rangle$ for $i = 1 \ldots, n$ where the norm $\|\cdot\|_n$ in $\mathbb{R}^n$ is the Euclidean norm multiplied by $1/\sqrt{n}$, it's adjoint operator $\mathrm{X}^* : \mathbb{R}^n \to \mathfrak{X}$ and the empirical covariance operator $\hat{\Sigma} = \mathrm{X}^*\,\mathrm{X}$, that are defined as

$$\mathrm{X}^*\,y = \frac{1}{n}\sum_{i=1}^{n} x_i y_i, \qquad \hat{\Sigma}w = \frac{1}{n}\sum_{i=1}^{n} \langle w, x_i \rangle\, x_i .$$

Similarly to the previous case $\mathrm{X}$ and $\hat{\Sigma}$ are bounded by $\kappa$ and $\kappa^2$ respectively.

From (22) it's easy to see that problem (2) can be rewritten as

$$\inf_{w \in \mathfrak{X}} \|Sw - f_\rho\|_{\rho_{\mathfrak{X}}}^2 .$$

Moreover, for the projection theorem it holds true that

$$S^* f_\rho = S^* f_{\mathcal{H}} ,$$

which implies that problem 2 can be rewritten as

$$\inf_{w \in \mathfrak{X}} \|Sw - f_{\mathcal{H}}\|_{\rho_{\mathfrak{X}}}^2 . \tag{23}$$

A regularization approach applied to the empirical risk minimization problem

$$\inf_{w \in \mathfrak{X}} \|\mathrm{X}\,w - \mathbf{y}\|_n^2 . \tag{24}$$

leads to an estimated solution of the form

$$\hat{w}_\lambda = g_\lambda(\hat{\Sigma})\,\mathrm{X}^*\,\mathbf{y} , \tag{25}$$

where $g_\lambda$ is a regularization function satisfying Definition 1 with qualification $q$ (Definition 2). Differently from the inverse problem setting we are trying to approximate a solution to the ideal problem (23) with a solution of the empirical problem (24) where $\mathrm{X}$, $\mathbf{y}$ are not only approximation of the ideal version $S$, $f_{\mathcal{H}}$ but are defined in different space.
Using the same regularization approach to the ideal problem we can define the unknown regularized solution as

$$w_\lambda = g_\lambda(\Sigma)S^* f_{\mathcal{H}} . \tag{26}$$

The performance of regularization algorithms $\hat{w}_\lambda$ can be measured in terms of the excess risk $\|S\hat{w}_\lambda - f_{\mathcal{H}}\|_{\rho_{\mathfrak{X}}}^2$. Assuming that $f_{\mathcal{H}} \in \mathcal{H}$, which implies that there exists some $w^*$ such that $f_{\mathcal{H}} = Sw^*$, it can be measured in terms of $\mathfrak{X}$-norm $\|\hat{w}_\lambda - w^*\|$ which is closely related to $\left\|L^{-1/2}S\,(\hat{w}_\lambda - w^*)\right\|_{\rho_{\mathfrak{X}}} = \left\|L^{-1/2}\,(S\hat{w}_\lambda - f_{\mathcal{H}})\right\|_{\rho_{\mathfrak{X}}}$ since for all $w \in \mathfrak{X}$

$$\left\|L^{-1/2}Sw\right\|_{\rho_{\mathfrak{X}}} \le \|w\| .$$

In what follows, we will measure the performance of algorithms in terms of a broader class of norms, $\|L^{-a}\,(S\hat{w}_\lambda - f_{\mathcal{H}})\|_{\rho_{\mathfrak{X}}}$, where $a \in [0, 1/2]$ is such that $L^{-a}f_{\mathcal{H}}$ is well defined.
Differently from the Assumption 4 here we consider a more general assumption on the target function and we don't assume any condition on the effective dimension $\mathcal{N}(\lambda)$.

**Assumption 5.**
*There exist $g_0 \in L_{\rho_{\mathfrak{X}}}^2$ such that*

$$f_{\mathcal{H}} = \Phi(L)g_0 , \quad \text{with } \|g_0\|_{\rho_{\mathfrak{X}}} \le R,$$

*where $\Phi : [0, \kappa^2] \to [0, +\infty)$ is a non-decreasing, operator monotone index function such that $\Phi(0) = 0$ and $\Phi(\kappa^2) < +\infty$. Moreover, for some $\zeta \in [0, q]$, the function $\Phi(u)u^{-\zeta}$ is non-decreasing, and the qualification $q$ of $g_\lambda$ covers the index function $\Phi$, which means that there exist a constant $c > 0$ such that for all $0 < \lambda < \kappa^2$,*

$$c\frac{\lambda^q}{\Phi(\lambda)} \le \inf_{\lambda \le u \le \kappa^2} \frac{u^q}{\Phi(u)} .$$

We are ready to state our general result.

**Theorem 4.**

*Under Assumption 2, 3, 5, let $\hat{w}_\lambda$ defined in (25), $a \in [0, 1/2]$, $\delta \in (0, 1/2)$ and $\lambda \in (0, 1)$. Assume the qualification $q$ of the chosen method to be larger than $1$ and the sample-size $n$ satisfy the following condition*

$$n \geq \frac{32\kappa^2\beta}{4\lambda} , \quad \beta = \log \frac{4\kappa^2(\mathcal{N}(\lambda) + 1)}{\delta \|\Sigma\|} ,$$

*then with probability at least $1 - \delta$ it holds true that*

$$\left\| L^{-a}(S\hat{w}_\lambda - f_{\mathcal{H}}) \right\|_{\rho_X} \leq \lambda^{-a} \left( \frac{\tilde{C}_1}{n\lambda^{\frac{1}{2}\vee(1-\zeta)}} + \left( \tilde{C}_2 + \frac{\tilde{C}_3}{\sqrt{n\lambda}} \right) \Phi(\lambda) + \tilde{C}_4 \sqrt{\frac{\mathcal{N}(\lambda)}{n}} \right) \log^2 \frac{2}{\delta}$$

*where the constants $\tilde{C}_1, \tilde{C}_2, \tilde{C}_3, \tilde{C}_4$ does not depend on $n, \lambda, \delta$.*
*In the follow we denote with $C$ a positive constant which does not depend on $n, \delta, \lambda$ and can be different every times it appears.*
*In particular, assuming $\lambda = O(n^{-\theta})$, and $n$ to be large enough, then with probability at least $1 - \delta$*

$$\left\| L^{-a}(S\hat{w}_\lambda - f_{\mathcal{H}}) \right\|_{\rho_X}^2 \leq C\lambda^{-2a} \left( \Phi(\lambda)^2 + \frac{\mathcal{N}(\lambda)}{n} \right) \log^2 \frac{2}{\delta} . \tag{27}$$

*Moreover assuming Holder source condition $\Phi(u) = u^r$ and that there exist $\gamma \geq 1, c_\gamma > 0$ such that $\mathcal{N}(\lambda) \leq c_\gamma \lambda^{-\frac{1}{\gamma}}$ then with probability at least $1 - \delta$ inequality 27 can be rewritten as*

$$\left\| L^{-a}(S\hat{w}_\lambda - f_{\mathcal{H}}) \right\|_{\rho_X}^2 \leq C\lambda^{-2a} \left( \lambda^{2r} + \frac{\lambda^{-\frac{1}{\gamma}}}{n} \right) \log^2 \frac{2}{\delta} , \tag{28}$$

*where if we choose $a = 0, \lambda = O(n^{-\frac{\gamma}{2\gamma r + 1}})$ we obtain the convergence result*

$$\left\| S\hat{w}_\lambda - f_{\mathcal{H}} \right\|_{\rho_X}^2 \leq C \left( n^{\frac{-2r\gamma}{2r\gamma + 1}} \right) \log^2 \frac{2}{\delta} . \tag{29}$$

## 8.3 Lemmas for Theorem 4

We firstly observe that Definition 1 and 2 of regularization function with qualification $q$ are equivalent to the following:

$$\sup_{\alpha \in [0,1]} \sup_{\lambda \in (0,1]} \sup_{u \in (0,\kappa^2]} |u^\alpha g_\lambda(u)| \leq E'\lambda^{1-\alpha} \tag{30}$$

$$\sup_{\alpha \in [0,q]} \sup_{\lambda \in (0,1]} \sup_{u \in (0,\kappa^2]]} |r_\lambda(u)u^\alpha\lambda^{-\alpha}| \leq F_q' .$$

where $E' = \max(E, F_0 + 1)$ and $F_q' = \max(F_0, F_q)$.

In this section we give some lemmas which are at the base of the proof of the learning bound.

### *Deterministic estimates*
The deterministic estimates concern the convergence term in the error bound

$$Sw_\lambda - f_{\mathcal{H}} = Sg_\lambda(\Sigma)S^* f_{\mathcal{H}} - f_{\mathcal{H}} = (g_\lambda(L)L - I)f_{\mathcal{H}} = -r_\lambda(L)f_{\mathcal{H}}$$

and it holds true the following lemma from [19].

**Lemma 2.**

*Under Assumption 4, let $w_\lambda$ given by (26), we have for all $a \in [0, \zeta]$,*

$$\left\| L^{-a}(Sw_\lambda - f_{\mathcal{H}}) \right\|_\rho \leq c_g R\Phi(\lambda)\lambda^{-a}, \quad c_g = \frac{F_q'}{c \wedge 1},$$

*and*

$$\|w_\lambda\| \leq E'\Phi(\kappa^2)\kappa^{-(2\zeta\wedge 1)}\lambda^{-(\frac{1}{2}-\zeta)_+} .$$

### *Probabilistic estimates*
Next lemma concern the concentration of the empirical mean of random variable in a Hilbert space.

**Lemma 3.**
*Let $w_1, \ldots, w_m$ be i.i.d. random variables in a Hilbert space with norm $\|\cdot\|$ and assume there exist two positive constants $B$ and $\sigma^2$ such that*

$$\mathbb{E}\left[\|w_1 - \mathbb{E}[w_1]\|^l\right] \leq \frac{1}{2}l!B^{l-2}\sigma^2, \quad \forall l \geq 2 .$$

*Then for any $0 < \delta < 1/2$, the following holds with probability at least $1 - \delta$,*

$$\left\|\frac{1}{m}\sum_{i=1}^{m} w_i - \mathbb{E}[w_i]\right\| \leq 2\left(\frac{B}{m} + \frac{\sigma}{\sqrt{m}}\right)\log\frac{2}{\delta} .$$

In the following two lemmas we control in probability the approximation of the covariance operator $\Sigma$ with the empirical covariance $\hat{\Sigma}$.

**Lemma 4.**
*Let $0 < \delta < 1/2$, it holds with probability at least $1 - \delta$:*

$$\left\|\Sigma - \hat{\Sigma}\right\|_{op} \leq \left\|\Sigma - \hat{\Sigma}\right\|_{HS} \leq \frac{6\kappa^2}{\sqrt{n}}\log\frac{2}{\delta} ,$$

*where $\|\cdot\|_{HS}$ denotes the Hilbert-Schmidt norm.*

This lemma is a consequence of the lemma above, (see e.g. [3] for a proof).

**Lemma 5.**
*Let $\delta \in (0, 1)$ and $\lambda > 0$. With probability at least $1 - \delta$ the following holds:*

$$\left\|\Sigma_\lambda^{-1/2}\left(\Sigma - \hat{\Sigma}\right)\hat{\Sigma}_\lambda^{-1/2}\right\|_{op} \leq \frac{4\kappa^2\beta}{3n\lambda} + \sqrt{\frac{2\kappa^2\beta}{n\lambda}} , \quad \beta = \log\frac{4\kappa^2\left(\mathcal{N}(\lambda) + 1\right)}{\delta\|\Sigma\|_{op}}$$

*where we denote $\Sigma_\lambda := \Sigma + \lambda\mathrm{I}$ and $\hat{\Sigma}_\lambda := \hat{\Sigma} + \lambda\mathrm{I}$.*

A proof of this result can be found in [17].

**Lemma 6.**
*Let $c, \delta \in (0, 1)$ and $\lambda > 0$. Assume*

$$n \geq \frac{32\kappa^2\beta}{(\sqrt{9 + 24c} - 3)^2\lambda} , \quad \beta = \log\frac{4\kappa^2\left(\mathcal{N}(\lambda) + 1\right)}{\delta\|\Sigma\|_{op}}$$

*then it holds with probability at least $1 - \delta$*

$$\left\|\Sigma_\lambda^{-1/2}\hat{\Sigma}_\lambda^{1/2}\right\|_{op}^2 \leq 1 + c$$

$$\left\|\Sigma_\lambda^{1/2}\hat{\Sigma}_\lambda^{-1/2}\right\|_{op}^2 \leq (1 - c)^{-1}$$

*In particular we will choose $c = 2/3$.*

*Proof.*
The condition

$$\frac{4\kappa^2\beta}{3n\lambda} + \sqrt{\frac{2\kappa^2\beta}{n\lambda}} \leq c$$

can be seen as a second degree inequality, and it's equivalent to the assumption

$$n \geq \frac{32\kappa^2\beta}{(\sqrt{9 + 24c} - 3)^2\lambda} , \quad \beta = \log\frac{4\kappa^2\left(\mathcal{N}(\lambda) + 1\right)}{\delta\|\Sigma\|} .$$

Applying Lemma 5 it holds true that

$$\left\|\Sigma_\lambda^{-1/2}\left(\Sigma - \hat{\Sigma}\right)\hat{\Sigma}_\lambda^{-1/2}\right\|_{op} \leq c .$$

Now, we can prove that

$$\left\|\Sigma_\lambda^{-1/2}\hat{\Sigma}_\lambda^{1/2}\right\|_{op}^2 = \left\|\Sigma_\lambda^{-1/2}\hat{\Sigma}_\lambda\hat{\Sigma}_\lambda^{-1/2}\right\|_{op} = \left\|\Sigma_\lambda^{-1/2}\left(\Sigma - \hat{\Sigma}\right)\hat{\Sigma}_\lambda^{-1/2} + I\right\|_{op}$$

$$\le \left\|\Sigma_\lambda^{-1/2}\left(\Sigma - \hat{\Sigma}\right)\hat{\Sigma}_\lambda^{-1/2}\right\|_{op} + \|I\|_{op} \le c + 1$$

which proves the first part of the thesis.
From [7] we get

$$\left\|\Sigma_\lambda^{1/2}\hat{\Sigma}_\lambda^{-1/2}\right\|_{op}^2 = \left\|\Sigma_\lambda^{1/2}\hat{\Sigma}_\lambda^{-1}\Sigma_\lambda^{1/2}\right\|_{op} = \left\|\left(I - \Sigma_\lambda^{-1/2}\left(\Sigma - \hat{\Sigma}\right)\hat{\Sigma}_\lambda^{-1/2}\right)^{-1}\right\|_{op} \le (1-c)^{-1}$$

which completes the proof. $\qquad\qquad\square$

The last important lemma regards the concentration of the empirical quantities $\hat{\Sigma}w_\lambda, X^*\mathbf{y}$ around the ideal ones $\Sigma w_\lambda, S^* f_{\mathcal{H}}$.

**Lemma 7.**
 *Under Assumptions 2, 3, 4, let $\delta \in (0, 1/2)$ and $w_\lambda$ given by (26), then the following holds with probability at least $1 - \delta$:*

$$\left\|\Sigma_\lambda^{-1/2}\left[\left(\hat{\Sigma}w_\lambda - X^*\mathbf{y}\right) - (\Sigma w_\lambda - S^* f_{\mathcal{H}})\right]\right\| \le \left(\frac{C_1}{n\lambda^{\frac{1}{2}\vee(1-\zeta)}} + \sqrt{\frac{C_2\Phi(\lambda)^2}{n\lambda} + \frac{C_3\mathcal{N}(\lambda)}{n}}\right)\log\frac{2}{\delta}$$

*where*

$$C_1 = 8\left(\kappa M + \kappa^2 E'\Phi(\kappa^2)\kappa^{-(2\zeta\wedge 1)}\right)$$
$$C_2 = 96c_g^2 R^2 \kappa^2$$
$$C_3 = 32(3B^2 + 4Q^2)\,.$$

*Proof.*
Let $\xi_i = \Sigma_\lambda^{-1/2}\left(\langle w, x_i\rangle - y_i\right)x_i$ for every $i \in 1, \ldots, n$, for the sake of simplicity we consider the random variable $\xi = \Sigma_\lambda^{-1/2}\left(\langle w, x\rangle - y\right)x$ independent and identically distributed to $\xi_i$ for every $i \in \{1, \ldots, n\}$. Now a simple calculation shows that

$$\mathbb{E}[\xi] = \Sigma_\lambda^{-1/2}\left(\Sigma w_\lambda - S^* f_{\mathcal{H}}\right)\,.$$

In order to apply Lemma 3, we bound $\mathbb{E}\left\|\xi - \mathbb{E}[\xi]\right\|^l$ for any $\mathbb{N} \ni l \ge 2$, in fact by using Holder inequality we get

$$\mathbb{E}\left\|\xi - \mathbb{E}[\xi]\right\|^l \le \mathbb{E}\left[|\|\xi\| - \mathbb{E}\|\xi\||\right]^l \le 2^{l-1}\left(\mathbb{E}\|\xi\|^l + (\mathbb{E}\|\xi\|)^l\right) \le 2^l \mathbb{E}\|\xi\|^l\,. \qquad (31)$$

We can proceed bounding

$$\mathbb{E}\|\xi\|^l = \mathbb{E}\left[\left\|\Sigma_\lambda^{-1/2}\left(y - \langle w_\lambda, x\rangle\right)x\right\|^l\right]$$

$$= \mathbb{E}\left[\left\|\Sigma_\lambda^{-1/2}x\right\|^l |y - \langle w_\lambda, x\rangle|^l\right]$$

$$\le 2^{l-1}\mathbb{E}\left[\left\|\Sigma_\lambda^{-1/2}x\right\|^l \left(|y|^l + |\langle w_\lambda, x\rangle|^l\right)\right]\,.$$

Now, thanks to (1) and Cauchy-Schwarz inequality, it holds that

$$\left\|\Sigma_\lambda^{-1/2}x\right\| \le \frac{\kappa}{\sqrt{\lambda}} \qquad (32)$$

$$|\langle w_\lambda, x\rangle| \le \|w_\lambda\|\|x\| \le \kappa\|w_\lambda\|\,. \qquad (33)$$

Thus we get, using again Cauchy-Schwarz inequality

$$\mathbb{E}\,\|\xi\|^l \leq 2^{l-1}\left(\frac{\kappa}{\sqrt{\lambda}}\right)^{l-2}\mathbb{E}\left[\left\|\Sigma_\lambda^{-1/2}x\right\|^2\left(|y|^l + (\kappa\,\|w_\lambda\|)^{l-2}\,|\langle w_\lambda, x\rangle\,|^2\right)\right]\ . \tag{34}$$

Regarding the first term of the sum, by Assumption 2,

$$\mathbb{E}\left[\left\|\Sigma_\lambda^{-1/2}x\right\|^2|y|^l\right] = \int_{\mathcal{X}}\left\|\Sigma_\lambda^{-1/2}x\right\|^2\int_{\mathbb{R}}|y|^l\,d\rho(y|x)\,d\rho_{\mathcal{X}}(x)$$

$$\leq \frac{1}{2}l!M^{l-2}Q^2\int_{\mathcal{X}}\left\|\Sigma_\lambda^{-1/2}x\right\|^2\,d\rho_{\mathcal{X}}(x)\ .$$

Observing that $\|w\| = \mathrm{Tr}\,(w\otimes w)$ it holds that

$$\int_{\mathcal{X}}\left\|\Sigma_\lambda^{-1/2}x\right\|^2\,d\rho_{\mathcal{X}}(x) = \int_{\mathcal{X}}\mathrm{Tr}\left(\Sigma_\lambda^{-1/2}x\otimes x\Sigma_\lambda^{-1/2}\right)\,d\rho_{\mathcal{X}}(x) = \mathrm{Tr}\left(\Sigma_\lambda^{-1/2}\Sigma\Sigma_\lambda^{-1/2}\right) = \mathcal{N}(\lambda)\ , \tag{35}$$

we get

$$\mathbb{E}\left[\left\|\Sigma_\lambda^{-1/2}x\right\|^2|y|^l\right] \leq \frac{1}{2}l!M^{l-2}Q^2\mathcal{N}(\lambda)\ . \tag{36}$$

Besides, Cauchy-Schwarz inequality implies that

$$\mathbb{E}\left[\left\|\Sigma_\lambda^{-1/2}x\right\|^2|\langle w_\lambda, x\rangle\,|^2\right] \leq 3\mathbb{E}\left[\left\|\Sigma_\lambda^{-1/2}x\right\|^2\left(|\langle w_\lambda, x\rangle - f_{\mathcal{H}}(x)|^2 + |f_{\mathcal{H}}(x) - f_\rho(x)|^2 + |f_\rho(x)|^2\right)\right]\ .$$

For the first term, Lemma 2 and inequality (32) implies that

$$\mathbb{E}\left[\left\|\Sigma_\lambda^{-1/2}x\right\|^2|\langle w_\lambda, x\rangle - f_{\mathcal{H}}(x)|^2\right] \leq \frac{\kappa^2}{\lambda}\mathbb{E}\left[|\langle w_\lambda, x\rangle - f_{\mathcal{H}}(x)^2|\right]$$

$$= \frac{\kappa^2}{\lambda}\|Sw_\lambda - f_{\mathcal{H}}\|_\rho^2$$

$$\leq c_g^2R^2\kappa^2\frac{\Phi(\lambda)^2}{\lambda}\ .$$

The second term can be controlled using Assumption 3,

$$\mathbb{E}\left[\left\|\Sigma_\lambda^{-1/2}x\right\|^2|f_{\mathcal{H}}(x) - f_\rho(x)|^2\right] = \mathbb{E}\left[\mathrm{Tr}\left(\Sigma_\lambda^{-1/2}x\otimes x\Sigma_\lambda^{-1/2}\right)(f_{\mathcal{H}}(x) - f_\rho(x))^2\right]$$

$$= \mathbb{E}\left[\mathrm{Tr}\left(\Sigma_\lambda^{-1}(f_{\mathcal{H}}(x) - f_\rho(x))^2x\otimes x\right)\right]$$

$$= \mathrm{Tr}\left(\Sigma_\lambda^{-1}\mathbb{E}\left[(f_{\mathcal{H}}(x) - f_\rho(x))^2x\otimes x\right]\right)$$

$$\leq B^2\,\mathrm{Tr}\left(\Sigma_\lambda^{-1}\Sigma\right) = B^2\mathcal{N}(\lambda)\ .$$

For the last term, by (12) and (35) we obtain

$$\mathbb{E}\left[\left\|\Sigma_\lambda^{-1/2}x\right\|^2|f_\rho(x)|^2\right] \leq Q^2\mathbb{E}\left[\left\|\Sigma_\lambda^{-1/2}x\right\|^2\right] = Q^2\mathcal{N}(\lambda)\ .$$

Therefore we obtain

$$\mathbb{E}\left[\left\|\Sigma_\lambda^{-1/2}x\right\|^2|\langle w_\lambda, x\rangle\,|^2\right] \leq 3\left(c_g^2R^2\kappa^2\frac{\Phi(\lambda)^2}{\lambda} + \left(B^2 + Q^2\right)\mathcal{N}(\lambda)\right)\ .$$

Now, putting this together with (36) in (34) we get

$$\mathbb{E}\,\|\xi\|^l \leq 2^{l-1}\left(\frac{\kappa}{\sqrt{\lambda}}\right)^{l-2}\left[\frac{1}{2}l!M^{l-2}Q^2\mathcal{N}(\lambda) + 3\,(\kappa\,\|w_\lambda\|)^{l-2}\left(c_g^2R^2\kappa^2\frac{\Phi(\lambda)^2}{\lambda} + \left(B^2 + Q^2\right)\mathcal{N}(\lambda)\right)\right]$$

$$\leq 2^{l-1}\frac{1}{2}l!\left(\frac{\kappa M + \kappa^2\,\|w_\lambda\|}{\sqrt{\lambda}}\right)^{l-2}\left(Q^2\mathcal{N}(\lambda) + 3\left(c_g^2R^2\kappa^2\frac{\Phi(\lambda)^2}{\lambda} + \left(B^2 + Q^2\right)\mathcal{N}(\lambda)\right)\right)$$

$$\leq 2^{l-1}\frac{1}{2}l!\left(\frac{\kappa M + \kappa^2\,\|w_\lambda\|}{\sqrt{\lambda}}\right)^{l-2}\left(3c_g^2R^2\kappa^2\frac{\Phi(\lambda)^2}{\lambda} + \left(3B^2 + 4Q^2\right)\mathcal{N}(\lambda)\right)\ .$$

Now by inequality (31) and Lemma 2 we obtain

$$\mathbb{E}\left\|\xi - \mathbb{E}\left[\xi\right]\right\|^l \leq \frac{1}{2}l!\left(\frac{4\left(\kappa M + \kappa^2 E'\Phi(\kappa^2)\kappa^{-(2\zeta\wedge 1)}\right)}{\lambda^{\frac{1}{2}\vee(1-\zeta)}}\right)^{l-2} 8\left(3c_g^2 R^2 \kappa^2 \frac{\Phi(\lambda)^2}{\lambda} + \left(3B^2 + 4Q^2\right)\mathcal{N}(\lambda)\right) .$$

The proof follows by applying Lemma 3. □

## *Operator inequalities*

**Lemma 8** ([12], Cordes inequalities)**.**
*Let $A, B$ be two positive bounded linear operators on a separable Hilbert space, then for all $s \in [0, 1]$*

$$\left\|A^s B^s\right\|_{op} \leq \left\|AB\right\|_{op}^s .$$

**Lemma 9** ([20, 21])**.**
*Let $\psi$ be an operator monotone index function on $[0, b]$, with $b > 1$. Then there is a constant $c_\psi < +\infty$ depending on $b - a$, such that for any pair $B_1, B_2$ such that $\left\|B_1\right\|_{op}, \left\|B_2\right\|_{op} \leq a$, of non-negative self-adjoint operators on some Hilbert space, it holds,*

$$\left\|\psi(B_1) - \psi(B_2)\right\| \leq c_\psi \psi\left(\left\|B_1 - B_2\right\|_{op}\right) .$$

*Moreover, there is $c'_\psi > 0$ such that*

$$c'_\psi \frac{\lambda}{\psi(\lambda)} \leq \frac{u}{\psi(u)}$$

*whenever $0 < \lambda < u \leq a \leq b$.*

### 8.4 Proof of Theorem 4

*Proof.*
It is a standard approach to decompose the error in the following way

$$\begin{aligned}
\left\|L^{-a}\left(S\hat{w}_\lambda - f_{\mathcal{H}}\right)\right\|_{\rho_X} &= \left\|L^{-a}\left[S\left(\hat{w}_\lambda - w_\lambda\right) + \left(Sw_\lambda - f_{\mathcal{H}}\right)\right]\right\|_{\rho_X} \\
&\leq \underbrace{\left\|L^{-a}S\left(\hat{w}_\lambda - w_\lambda\right)\right\|_{\rho_X}}_{stability} + \underbrace{\left\|L^{-a}\left(Sw_\lambda - f_{\mathcal{H}}\right)\right\|_{\rho_X}}_{convergence} .
\end{aligned} \quad (37)$$

With this error decomposition the first term of the sum depends on how much the empirical and ideal problem are related, while the second term depends on the convergence properties of the regularization method used.

## *Convergence*
Lemma 2 implies that the convergence term can be controlled with

$$\left\|L^{-a}\left(Sw_\lambda - f_{\mathcal{H}}\right)\right\|_{\rho_X} \leq c_g R\Phi(\lambda)\lambda^{-a} . \quad (38)$$

## *Stability*
Regarding the stability term we first observe that by Lemma 4, 6 (with $c = 2/3$) and 7 and assuming

$$n \geq \frac{32\kappa^2\beta}{4\lambda} , \quad \beta = \log\frac{4\kappa^2\left(\mathcal{N}(\lambda) + 1\right)}{\delta\left\|\Sigma\right\|}$$

then with probability at least $1 - \delta$ it holds true that

$$\left\|\Sigma_\lambda^{-1/2}\hat{\Sigma}_\lambda^{1/2}\right\|_{op}^2 \vee \left\|\Sigma_\lambda^{1/2}\hat{\Sigma}_\lambda^{-1/2}\right\|_{op}^2 \leq \Delta_1$$

$$\left\|\Sigma_\lambda^{-1/2}\left[\left(\hat{\Sigma}w_\lambda - X^* \mathbf{y}\right) - \left(\Sigma w_\lambda - S^* f_{\mathcal{H}}\right)\right]\right\| \leq \Delta_2$$

$$\left\|\Sigma - \hat{\Sigma}\right\|_{op} \leq \left\|\Sigma - \hat{\Sigma}\right\|_{HS} \leq \Delta_3$$

where

$$\Delta_1 = 3$$

$$\Delta_2 = \left( \frac{C_1}{n\lambda^{\frac{1}{2}\vee(1-\zeta)}} + \sqrt{\frac{C_2\Phi(\lambda)^2}{n\lambda} + \frac{C_3\mathcal{N}(\lambda)}{n}} \right) \log\frac{2}{\delta}$$

$$\Delta_3 = \frac{6\kappa^2}{\sqrt{n}} \log\frac{2}{\delta} \ .$$

We now begin with the following inequality

$$\left\| L^{-a}S\left(\hat{w}_\lambda - w_\lambda\right) \right\|_{\rho_X} \leq \left\| L^{-a}S\Sigma_\lambda^{a-\frac{1}{2}} \right\|_{op} \left\| \Sigma_\lambda^{\frac{1}{2}-a}\hat{\Sigma}_\lambda^{a-\frac{1}{2}} \right\|_{op} \left\| \hat{\Sigma}_\lambda^{\frac{1}{2}-a}(\hat{w}_\lambda - w_\lambda) \right\|$$

where, thanks to spectral theorem and Cordes inequality, the first two terms can be controlled as follows:

$$\left\| L^{-a}S\Sigma_\lambda^{a-1/2} \right\|_{op} \leq \left\| L^{-a}S\Sigma^{a-\frac{1}{2}} \right\|_{op} \leq 1$$

$$\left\| \Sigma_\lambda^{\frac{1}{2}-a}\hat{\Sigma}_\lambda^{a-1/2} \right\|_{op} = \left\| \Sigma_\lambda^{\frac{1}{2}(1-2a)}\hat{\Sigma}_\lambda^{-\frac{1}{2}(1-2a)} \right\|_{op} \leq \Delta_1^{\frac{1}{2}-a} \ .$$

Now adding and subtracting the mixed-term $\hat{\Sigma}_\lambda^{\frac{1}{2}-a}g_\lambda(\hat{\Sigma})\hat{\Sigma}w_\lambda$ and using triangular inequality we obtain

$$\left\| L^{-a}S\left(\hat{w}_\lambda - w_\lambda\right) \right\|_{\rho_X} \leq \Delta_1^{\frac{1}{2}-a} \left( \left\| \hat{\Sigma}_\lambda^{\frac{1}{2}-a}\left( \hat{w}_\lambda - g_\lambda(\hat{\Sigma})\hat{\Sigma}w_\lambda \right) \right\| + \left\| \hat{\Sigma}_\lambda^{\frac{1}{2}-a}r_\lambda(\hat{\Sigma})w_\lambda \right\| \right)$$

$$= \Delta_1^{\frac{1}{2}-a} \left( \left\| \hat{\Sigma}_\lambda^{\frac{1}{2}-a}g_\lambda(\hat{\Sigma})\left( \mathrm{X}^*\mathbf{y} - \hat{\Sigma}w_\lambda \right) \right\| + \left\| \hat{\Sigma}_\lambda^{\frac{1}{2}-a}r_\lambda(\hat{\Sigma})w_\lambda \right\| \right) \ . \quad (39)$$

**Estimating** $\left\| \hat{\Sigma}_\lambda^{\frac{1}{2}-a}g_\lambda(\hat{\Sigma})\left( \mathrm{X}^*\mathbf{y} - \hat{\Sigma}w_\lambda \right) \right\|$ **:**

We first have

$$\left\| \hat{\Sigma}_\lambda^{\frac{1}{2}-a}g_\lambda(\hat{\Sigma})\left( \mathrm{X}^*\mathbf{y} - \hat{\Sigma}w_\lambda \right) \right\| \leq \left\| \hat{\Sigma}_\lambda^{\frac{1}{2}-a}g_\lambda(\hat{\Sigma})\hat{\Sigma}_\lambda^{\frac{1}{2}} \right\|_{op} \left\| \hat{\Sigma}_\lambda^{-\frac{1}{2}}\Sigma_\lambda^{\frac{1}{2}} \right\|_{op} \left\| \Sigma_\lambda^{-\frac{1}{2}}\left( \mathrm{X}^*\mathbf{y} - \hat{\Sigma}w_\lambda \right) \right\| \ .$$

Now, thanks to the definition of regularization function $g_\lambda$ and since $\hat{\Sigma}$ is bounded by $\kappa^2$

$$\left\| \hat{\Sigma}_\lambda^{\frac{1}{2}-a}g_\lambda(\hat{\Sigma})\hat{\Sigma}_\lambda^{\frac{1}{2}} \right\|_{op} \leq \sup_{u\in[0,\kappa^2]} |(u+\lambda)^{1-a}g_\lambda(u)|$$

$$\leq \sup_{u\in[0,\kappa^2]} |(u^{1-a}+\lambda^{1-a})g_\lambda(u)|$$

$$\leq 2E'\lambda^{-a} \ .$$

Thus we obtain

$$\left\| \hat{\Sigma}_\lambda^{\frac{1}{2}-a}g_\lambda(\hat{\Sigma})\left( \mathrm{X}^*\mathbf{y} - \hat{\Sigma}w_\lambda \right) \right\| \leq 2E'\lambda^{-a}\Delta_1^{\frac{1}{2}} \left\| \Sigma_\lambda^{-\frac{1}{2}}\left( \hat{\Sigma}w_\lambda - \mathrm{X}^*\mathbf{y} \right) \right\| \ .$$

Now, adding and subtracting $\Sigma_\lambda^{-\frac{1}{2}}\left( \Sigma w_\lambda - S^*f_{\mathcal{H}} \right)$ we obtain

$$\left\| \Sigma_\lambda^{-\frac{1}{2}}\left( \hat{\Sigma}w_\lambda - \mathrm{X}^*\mathbf{y} \right) \right\| \leq \left\| \Sigma_\lambda^{-\frac{1}{2}}\left[ \left( \hat{\Sigma}w_\lambda - \mathrm{X}^*\mathbf{y} \right) - \left( \Sigma w_\lambda - S^*f_{\mathcal{H}} \right) \right] \right\| + \left\| \Sigma_\lambda^{-\frac{1}{2}}\left( \Sigma w_\lambda - S^*f_{\mathcal{H}} \right) \right\|$$

$$\leq \left\| \Sigma_\lambda^{-\frac{1}{2}}\left[ \left( \hat{\Sigma}w_\lambda - \mathrm{X}^*\mathbf{y} \right) - \left( \Sigma w_\lambda - S^*f_{\mathcal{H}} \right) \right] \right\| + \left\| \Sigma_\lambda^{-\frac{1}{2}}S^* \right\|_{op} \| Sw_\lambda - f_{\mathcal{H}} \|$$

$$\leq \Delta_2 + c_g R\Phi(\lambda) \ ,$$

where in the last inequality we use Lemma 2 and that $\left\| \Sigma_\lambda^{-\frac{1}{2}}S^* \right\| \leq 1$. We thus obtain that

$$\left\| \hat{\Sigma}_\lambda^{\frac{1}{2}-a}g_\lambda(\hat{\Sigma})\left( \mathrm{X}^*\mathbf{y} - \hat{\Sigma}w_\lambda \right) \right\| \leq 2E'\lambda^{-a}\Delta_1^{\frac{1}{2}}\left( \Delta_2 + c_g R\Phi(\lambda) \right) \ . \quad (40)$$

**Estimating** $\left\|\hat{\Sigma}_\lambda^{\frac{1}{2}-a} r_\lambda(\hat{\Sigma}) w_\lambda\right\|$ **:**

Note that from the definition of $w_\lambda$ it holds that

$$w_\lambda = g_\lambda(\Sigma) S^* \Phi(L) g_0 = g_\lambda(\Sigma) \Phi(\Sigma) S^* g_0 \ ,$$

and thus,

$$\left\|\hat{\Sigma}_\lambda^{\frac{1}{2}-a} r_\lambda(\hat{\Sigma}) w_\lambda\right\| \leq R \left\|\hat{\Sigma}_\lambda^{\frac{1}{2}-a} r_\lambda(\hat{\Sigma}) g_\lambda(\Sigma) \Phi(\Sigma) S^*\right\|_{op} = R \left\|\hat{\Sigma}_\lambda^{\frac{1}{2}-a} r_\lambda(\hat{\Sigma}) g_\lambda(\Sigma) \Phi(\Sigma) \Sigma^{\frac{1}{2}}\right\|_{op} \ .$$

Now we have

$$\left\|\hat{\Sigma}_\lambda^{\frac{1}{2}-a} r_\lambda(\hat{\Sigma}) g_\lambda(\Sigma) \Phi(\Sigma) \Sigma^{\frac{1}{2}}\right\|_{op} \leq \left\|\hat{\Sigma}_\lambda^{\frac{1}{2}-a} r_\lambda(\hat{\Sigma}) \hat{\Sigma}_\lambda^{\frac{1}{2}}\right\|_{op} \left\|\hat{\Sigma}_\lambda^{-\frac{1}{2}} \Sigma_\lambda^{\frac{1}{2}}\right\|_{op} \left\|\Sigma_\lambda^{-\frac{1}{2}} \Sigma^{\frac{1}{2}}\right\|_{op} \|g_\lambda(\Sigma) \Phi(\Sigma)\|_{op}$$

$$\leq \Delta_1^{\frac{1}{2}} \left\|\hat{\Sigma}_\lambda^{1-a} r_\lambda(\hat{\Sigma})\right\|_{op} \|g_\lambda(\Sigma) \Phi(\Sigma)\|_{op} \ .$$

For the first term we get

$$\left\|\hat{\Sigma}_\lambda^{1-a} r_\lambda(\hat{\Sigma})\right\| \leq \sup_{u \in [0, \kappa^2]} |(u + \lambda)^{1-a} r_\lambda(u)| \leq 2 F_q' \lambda^{1-a} \ ,$$

while for the second term we have

$$\|g_\lambda(\Sigma) \Phi(\Sigma)\|_{op} \leq \sup_{u \in [0, \kappa^2]} |g_\lambda(u) \Phi(u)| \ .$$

Now if $0 < u \leq \lambda$, as $\Phi(u)$ is non-decreasing, $\Phi(u) \leq \Phi(\lambda)$, hence by (9) we obtain

$$g_\lambda(u) \Phi(u) \leq E' \Phi(\lambda) \lambda^{-1} \ .$$

When $\lambda \leq u \leq \kappa^2$, following from Lemma 9, there is a constant $c_\Phi' \geq 1$ such that

$$\Phi(u) u^{-1} \leq c_\Phi' \Phi(\lambda) \lambda^{-1} \ ,$$

thus, by (9), we get

$$g_\lambda(u) \Phi(u) = g_\lambda(u) u \Phi(u) u^{-1} \leq E' c_\Phi' \Phi(\lambda) \lambda^{-1} \ .$$

Therefore for all $0 < u \leq \kappa^2$, $g_\lambda(u) \Phi(u) \leq E' c_\Phi' \Phi(\lambda) \lambda^{-1}$ and we can conclude that

$$\left\|\hat{\Sigma}_\lambda^{\frac{1}{2}-a} r_\lambda(\hat{\Sigma}) w_\lambda\right\|_{op} \leq 2 R \Delta_1^{\frac{1}{2}} F_q' E' c_\Phi' \Phi(\lambda) \lambda^{-a} \ . \tag{41}$$

## *Learning bounds*

We are now ready to state the learning bound related to the regularized solution $\hat{w}_\lambda = g_\lambda(\hat{\Sigma}) X^* \mathbf{y}$. By combining (37), (38), (39), (40), (41) we obtain that with probability at least $1 - \delta$

$$\begin{aligned}
\left\|L^{-a} (S\hat{w}_\lambda - f_{\mathcal{H}})\right\|_{\rho_{\mathcal{X}}} \leq \ & c_g R \Phi(\lambda) \lambda^{-a} \\
& + \Delta_1^{1-a} 2 E' \left(\Delta_2 + c_g R \Phi(\lambda)\right) \lambda^{-a} \\
& + \Delta_1^{1-a} 2 R F_q' E' c_\Phi' \Phi(\lambda) \lambda^{-a} \ .
\end{aligned}$$

Since

$$\Delta_2 \leq \left(\frac{C_1}{n \lambda^{\frac{1}{2} \vee (1-\zeta)}} + \sqrt{\frac{C_2 \Phi(\lambda)^2}{n\lambda}} + \sqrt{\frac{C_3 \mathcal{N}(\lambda)}{n}}\right) \log \frac{2}{\delta}$$

and $\log \frac{2}{\delta} \geq 1$ for all $\delta \in (0, 1/2)$, we can rewrite the above inequality as

$$\left\|L^{-a} (S\hat{w}_\lambda - f_{\mathcal{H}})\right\|_{\rho_{\mathcal{X}}} \leq \lambda^{-a} \left(\frac{\tilde{C}_1}{n \lambda^{\frac{1}{2} \vee (1-\zeta)}} + \left(\tilde{C}_2 + \frac{\tilde{C}_3}{\sqrt{n\lambda}}\right) \Phi(\lambda) + \tilde{C}_4 \sqrt{\frac{\mathcal{N}(\lambda)}{n}}\right) \log \frac{2}{\delta}$$

where

$$\begin{aligned}
\tilde{C}_1 &= 2\Delta_1^{1-a} E' C_1 = 16 \cdot 3^{1-a} E' \left(\kappa M + \kappa^2 E' \Phi(\kappa^2) \kappa^{-(2\zeta \wedge 1)}\right) \\
\tilde{C}_2 &= c_g R + 2\Delta_1^{1-a} E' R \left(c_g + c_\Phi' F_q'\right) \\
\tilde{C}_3 &= 2\sqrt{C_2} \Delta_1^{1-a} E' = 2\sqrt{96 c_g^2 R^2 \kappa^2} 3^{1-a} E' \\
\tilde{C}_4 &= 2\sqrt{C_3} \Delta_1^{1-a} E' = 2\sqrt{32(3B^2 + 4Q^2)} 3^{1-a} E' \ .
\end{aligned}$$

which complete the proof of the first part of the thesis.
Computing the square of the previous we have

$$\left\| L^{-a} \left( S\hat{w}_\lambda - f_{\mathcal{H}} \right) \right\|_{\rho_x}^2 \leq 3\lambda^{-2a} \left[ \left( \frac{\tilde{C}_1}{n\lambda^{\frac{1}{2} \vee (1-\zeta)}} \right)^2 + \left( \tilde{C}_2 + \frac{\tilde{C}_3}{\sqrt{n\lambda}} \right)^2 \Phi(\lambda)^2 + \left( \tilde{C}_4 \sqrt{\frac{\mathcal{N}(\lambda)}{n}} \right)^2 \right] \log^2 \frac{2}{\delta} \ .$$

Assuming $\lambda$ to be of the order $O\left( n^{-\theta} \right)$, for some $\theta \in (0, 1)$, then

$$\lim_{n \to \infty} \frac{1}{n\lambda} = 0 \quad , \quad \lim_{n \to \infty} \Phi(\lambda) = 0 \ ,$$

thus, assuming $n$ to be large enough, we can ignore the second order terms, hence we have that for some positive constant $C$ which does not depend on $n, \lambda, \delta$, it holds true that

$$\left\| L^{-a} \left( S\hat{w}_\lambda - f_{\mathcal{H}} \right) \right\|_{\rho_x}^2 \leq C\lambda^{-2a} \left( \Phi(\lambda)^2 + \frac{\mathcal{N}(\lambda)}{n} \right) \log^2 \frac{2}{\delta} \ . \tag{42}$$

Now if we assume Holder condition $\Phi(u) = u^r$ and $\mathcal{N}(\lambda) \leq c_\gamma \lambda^{-\frac{1}{\gamma}}$ then (42) implies

$$\left\| L^{-a} \left( S\hat{w}_\lambda - f_{\mathcal{H}} \right) \right\|_{\rho_x}^2 \leq C\lambda^{-2a} \left( \lambda^{2r} + \frac{\lambda^{-\frac{1}{\gamma}}}{n} \right) \log^2 \frac{2}{\delta} \ .$$

By balancing the two terms

$$\lambda^{2r} = \frac{\lambda^{-\frac{1}{\gamma}}}{n}$$

we get the choice for the regularization parameter

$$\lambda = O(n^{-\frac{\gamma}{2\gamma r + 1}})$$

which in the case $a = 0$ directly implies (29).

$\square$

## 8.5 Proof of Theorem 1 and 2

*Proof.*
Both Theorem 1 and 2 follow from Theorem 4 by choosing $\lambda = \frac{1}{t}$ for gradient descent and $\lambda = \frac{1}{t^2}$ for accelerated methods.
However in the estimation of the term $\left\| \hat{\Sigma}_\lambda^{\frac{1}{2} - a} r_\lambda(\hat{\Sigma}) w_\lambda \right\|$ of the stability (39) we use the the fact that the qualification of the method is at least 1. We can obtain the same result for Nesterov method by assuming furthermore that the parameter $r$ of the source condition to be larger than $1/2$. We have

$$\left\| \hat{\Sigma}_\lambda^{\frac{1}{2}} r_\lambda(\hat{\Sigma}) w_\lambda \right\| \leq \left\| \hat{\Sigma}_\lambda^{\frac{1}{2}} r_\lambda(\hat{\Sigma}) \right\|_{op} \|w_\lambda\| = \left\| \left( \hat{\Sigma} + \lambda \mathrm{I} \right)^{\frac{1}{2}} r_\lambda(\hat{\Sigma}) \right\|_{op} \|w_\lambda\| \ .$$

Thanks to the source condition we have that the norm $w_\lambda$ is bounded

$$\|w_\lambda\| = \|g_\lambda(\Sigma)\Sigma^r S^* g_0\| \leq \left\| g_\lambda(\Sigma)\Sigma^{r+\frac{1}{2}} \right\|_{op} \|g_0\| \leq \kappa^{2r-1} E' \|g_0\| \ .$$

On the other hand

$$\left\| \hat{\Sigma}_\lambda^{\frac{1}{2}} r_\lambda(\hat{\Sigma}) \right\|_{op} \leq \sup_{u \in [0, \kappa^2]} \left| \left( \sqrt{u + \lambda} \right) r_\lambda(u) \right| \leq 2F'_q \lambda^{1/2} \ .$$

This complete the proof.

$\square$