[Reviews · NeurIPS 2019]

Reviewer 1



The authors studied the implicit regularization effect of (accelerated) gradient algorithms for least squares. The paper is well-written and clear. The main results are the population risk analysis of (accelerated) gradient methods for least squares, with the conclusion that accelerated methods achieve the same accuracy of vanilla gradient descent but with much fewer iterations. The assumptions are clean and standard. I like that the convergence bounds are proved in a very "unified" way based on spectral filtering technique. The authors also showed that their theoretical findings are empirically observed (and matched almost perfectly!) in the experiments, which is very nice. One drawback of the paper is that there are only upper bounds without any corresponding lower bounds. In other words, it is possible that their analysis are not tight, although that seems unlikely to me according to the experiments. Also, the least square setting is a bit too simplified (is it possible to extend to general convex setting?), but I am OK with just least squares at the current state of research. Overall, I think this is a good theory paper which connects stats and optimization. I suggest acceptance.

Reviewer 2



**********After author response*********** I thank the authors for answering my questions. In particular, my concern about novelty was addressed. I decide to raise my score. ************************************************ Originality: There are many previous results about the spectral filtering functions of AGD and the authors are not very clear about what's novel. See entry 1 in the "improvements" part. Quality: I haven't checked the proofs in the appendices, but the theoretical insight is convincingly verified by experiments. Clarity: The paper is clearly written in general, although there are minor issues that I mention in entries 2, 3 and 4 in the "improvements" part. Significance: The generalization bounds for AGD in this work will enhance the statistical understanding of optimization methods. While there are previous works showing that AGD is less stable than GD, the finding that AGD achieves the same optimal excess risk as GD is interesting.

Reviewer 3



The paper reviews Gradient Decent and accelerated methods and provide novel analysis for the accelerated methods. Originality - the analysis techniques using in the paper are novel. The paper combines known techniques to conclude improved analysis for the known methods. The related work is very well cited and explained. Clarity. The paper is very well written and organized. Very good review of the subject. The technical parts were left to the appendix. The theorems in the paper are a direct conclusion from "Theorem 4" which is not stated in the main part of the paper. The appendix is harder to understand, The mathematical analysis seems to be correct. Quality - The authors made a complete work, providing claims analysis and empirical demonstrations. Most of the work is provided in the appendix. The paper also provided strength and weaknesses for their work. A further analysis of the unstable behavior would be interesting. Significance. The paper set interesting starting point for further study. It is hard to evaluate the significance of this paper. The analysis techniques are novel and led to improved analysis for the problem. But the paper does not focus on the techniques or how to use them in other scenarios. The results are also novel, but also shown to be "unstable", as the test error tend to explode past the minimum point. It is unclear from practical point of view when should accelerated method should be used.

[Author Response · NeurIPS 2019]

1 We would like to thank all the reviewers for all the suggestive comments. Since all of the reviewers have different
2 questions and doubts we decide to answer individually. Because of a lack of space we cite the papers using the
3 references of the submitted work.

**Reviewer #1**:
**1)** Only upper bound! In this paper we only present upper bounds, but we also observe that our rate of convergence is
indeed optimal since it matches corresponding lower bounds [3,7]. We will stress this important fact.
**2)** Extend the result to general convex loss function! Yes, we analyze the least-squares setting because of the easier
structure. We expect similar results could hold for other loss (e.g. convex-Lipshitz, or even just smooth twice
differentiable), but proofs, as well as details, to be different.

**Reviewer #2**:
**1)** What's new and what's known? The filtering properties of $\nu$-method and Nesterov accelerated algorithm for inverse
problems have been studied respectively in [10] and [22]. We use these results together with probabilistic tools to derive
their learning properties. Concerning the novelty in Theorem 1, implicit regularization properties of gradient descent
algorithm have been largely studied both in the learning context (see e.g. [30,6]) and in the inverse problem scenario
(see e.g. [10]). For what concern the optimization properties of accelerated methods, they have been studied since
[22,10] but only in inverse problems.
**2)** Do the constants $C_1$ and $C_2$ depends on $R$ and $c_\gamma$? Yes, they do, the dependence can be tracked in the proof of the
general result and is at most linear for both $R$ and $c_\gamma$. We will ass a comment on this point.
**3)** How the qualification affects the learning bound? Theorem 2 holds if the parameter $r$ of the source condition is
smaller than the qualification parameter of the chosen optimization algorithm, hence a higher qualification can adapt to
better properties of the unknown target function. In this paper, we didn't focus on the effects of the qualification, but we
will add some plots to illustrate this. See below.

In these simulations (where we chose the same parameters of those in the paper) it can be observed that increasing the parameter $r$ of the source condition gradient descent (qualification $\infty$) can recover the behavior of the other methods.

**4)** Infimum over $\mathcal{H}$? It's a typos, the infimum is over $\mathcal{X}$, but extending the expected risk to $L^2(\mathcal{X}, \rho_\mathcal{X})$ and denoting
with $\mathcal{H}$ the subspace of linear function then they are the same.

**Reviewer #3**:
**1)** Advantages of accelerated methods? Considering the large-scale scenario or problems of learning on a budget, one
cannot chose to pay with time, for example if the optimal iterations are $10^6$ for GD and $10^3$ for the accelerated versions
but we have a limited budget of $10^2$ iterations then acceleration is to prefer because of the faster decrease. Accelerated
methods allow to reach the same accuracy with less computation, so they are to prefer. It is true that in practice it is
difficult to stop at the optimal iteration and the plateau of "good iterations" is more strict for the accelerated methods,
but this problem concern how to tune hyperparameters not the algorithm itself. On the other hand gradient descent can
exploit it's higher qualification (see point 3 of reviewer #2), but again in practice one do not know nothing about source
condition.
**2)** Accelerated methods are more unstable! We call "stability" the error which derive from running the algorithm with
finite data instead of infinite, it can be seen at line 434 of Supplementary material that this term turns out to be the
second addend of the learning bound. For accelerated methods this term is the square of the gradient descent one, so
accelerated methods are more unstable.
**3)** Why placing $1/t$ and $1/t^2$? Main mathematical contribution in the appendix! The proof holds in general for all
spectral algorithms defined by a filter function $g_\lambda$. In Section 3 we prove that gradient descent and the accelerated
methods are filter functions by choosing the regularization parameter $\lambda$ respectively as $1/t$ and $1/t^2$. Yes, the theorem
in the appendix is more general but in this paper we wanted to focus on the statistical properties of the acceleration
rather that general spectral filtering.
**4)** Missing definitions! It's true, the function $g_t$ can be extended through spectral calculus to a function of operators by
defining it on the eigenvalues. The operator $x \otimes x$ is defined as the operator form $\mathcal{X}$ to $\mathcal{X}$ such that $x \otimes x(v) = \langle x, v \rangle x$,
and $\Sigma = \mathbb{E}[x \otimes x]$.

We thank again the reviewers and hope to have clarified all their doubts.

[Meta-Review · NeurIPS 2019]

This paper investigates Nesterov's acceleration technique and early stopping problem for optimization in RKHS for a nonparametric regression problem. It is shown that the acceleration technique induces earlier optimal stopping time to achieve the optimal generalization error (minimax optimal rate). This paper deals with an interesting problem and the analysis is novel. The result is interesting because the acceleration technique might cause instability but the result of this paper gives a proper answer to this issue. The analysis given in this paper will stimulate further researches in that direction.